# Control of CRK-RAC1 activity by the *miR-1/206/133* miRNA family is essential for neuromuscular junction function

Ina Klockner[1], Christian Schutt[1], Theresa Gerhardt[1], Thomas Boettger [1✉] & Thomas Braun [1✉]

Formation and maintenance of neuromuscular junctions (NMJs) are essential for skeletal muscle function, allowing voluntary movements and maintenance of the muscle tone, thereby preventing atrophy. Generation of NMJs depends on the interaction of motor neurons with skeletal muscle fibers, which initiates a cascade of regulatory events that is essential for patterning of acetylcholine receptor (AChR) clusters at specific sites of the sarcolemma. Here, we show that muscle-specific miRNAs of the *miR-1/206/133* family are crucial regulators of a signaling cascade comprising DOK7-CRK-RAC1, which is critical for stabilization and anchoring of postsynaptic AChRs during NMJ development and maintenance. We describe that posttranscriptional repression of CRK by *miR-1/206/133* is essential for balanced activation of RAC1. Failure to adjust RAC1 activity severely compromises NMJ function, causing respiratory failure in neonates and neuromuscular symptoms in adult mice. We conclude that *miR-1/206/133* serve a specific function for NMJs but are dispensable for skeletal muscle development.

[1] Max Planck Institute for Heart- and Lung Research, Department of Cardiac Development and Remodelling, Ludwigstr. 43, D-61231 Bad Nauheim, Germany. ✉email: thomas.boettger@mpi-bn.mpg.de; thomas.braun@mpi-bn.mpg.de

Neuromuscular junctions (NMJs) are chemical synapsis, critical for transmitting action potentials from motor neurons to skeletal muscle fibers and initiation of sarcolemma depolarization, eventually resulting in $Ca^{2+}$ release and muscle contraction. The function of NMJs is severely impaired in several neuromuscular diseases such as Congenital Myasthenic Syndromes (CMS), Amyotrophic Lateral Sclerosis (ALS), and Myasthenia Gravis (MG), entailing compromised muscle function, locomotion, and survival[1]. Development and maintenance of NMJs require the interaction between motor neurons, terminal Schwann cells, and myofibers, eventually leading to clustering of acetylcholine receptors (AChR), a hallmark of NMJs. Formation of NMJs is initiated by a signaling cascade, involving the receptor tyrosine kinase MuSK, its co-receptor LRP4, and the intracellular Docking Protein 7 (DOK7)[2,3]. Binding of the presynaptically secreted proteoglycan agrin to postsynaptic LRP4 stimulates MuSK dimerization, phosphorylation, and recruitment of DOK7, thereby sustaining MuSK activity, but also recruiting the adapter proteins CRK and CRK-L[4,5]. Mutation of *Dok7* (1124_1127dupTGCC), which abolishes binding of CRK/CRK-L to DOK7, causes CMS[6,7]. Although it was found that expression of both CRK and CRK-L is essential for AChR clustering during NMJ formation[4,8], the events downstream of CRK for NMJ formation are not well understood. It has been speculated that CRK might interact with guanine nucleotide exchange factors (GEFs) at the NMJ to stimulate activation of small GTPases but experimental evidence for this hypothesis is lacking[4,9].

miRNAs control gene expression at the posttranscriptional level, either by inducing degradation of target mRNAs or by inhibiting their translation[10]. Furthermore, miRNAs present in the serum represent useful biomarkers and may serve additional functions in interorgan signaling[11]. The miRNAs of the *miR-1/206/133* gene family belong to the most abundant miRNAs in striated muscles and are expressed from three separate genomic microRNA clusters. *miR-1/206/133* miRNAs fulfill distinct functions at different developmental stages in the heart as well as in skeletal muscle, which may depend on differential expression of target genes, blockage of putative binding sites, or other reasons[12–15]. *miR-1* and *miR-133a* are transcribed from two different gene clusters as bi-cistronic pri-miRNA molecules, which are processed to mature *miR-1* and *miR-133a* molecules. *miR-1* and *miR-133a* are co-expressed, since they are derived from a joint pri-miRNA and cooperate to regulate related physiological functions. Deletion of the two genomic *miR-1/133a* clusters revealed a synergistic function of *miR-1/133a* in repressing myocardin and other components of the smooth muscle gene program during early cardiac development[12]. Similarly, concomitant deletion of *miR-1-1* and *miR-1-2* activates the smooth muscle gene program in cardiomyocytes and compromises sarcomere formation but does not result in embryonic lethality, confirming synergistic functions of *miR-1* and *miR-133a*[13]. Inactivation of *miR-133a* causes developmental abnormalities due to aberrant cell proliferation and increased smooth muscle gene expression[16]. In adult skeletal muscles, *miR-1/133a* is required for metabolic maturation by suppressing the *Dlk1-Dio3* miRNA/lncRNA mega-cluster via MEF2A[14]. Surprisingly, however, inactivation of the two *miR-1/133a* gene clusters had no impact on embryonic and postnatal skeletal muscle development, which might be due to compensatory effects of the related *miR-206/133b* miRNA cluster[17] that is specifically expressed in skeletal muscles but not the heart.

Here, we demonstrate that the miRNAs of the *miR-1/206/133* family synergize to play a crucial role in the formation and maintenance of the neuromuscular junction but are dispensable for skeletal muscle development. We discovered that expression of CRK, which is essential for formation and maintenance of

NMJs in skeletal muscles, is adjusted by the concomitant activity of *miR-1/206/133*. Increased expression of CRK enhances its interaction with RAC1-GEF FARP1, thereby boosting activation of the small GTPase RAC1, which prevents formation and maintenance of AChR macro-clusters, essential for NMJ function.

## Results

**miR-206/133b compensate for miR-1/133a in skeletal muscle.** The three miRNA *miR-1/206/133* gene clusters are highly expressed in skeletal muscle[18,19]. During skeletal muscle development and maturation, the expression of *miR-1*, *miR-133*, and *miR-206* continuously increases, reaching highest levels in adult myofibers. This also holds true for *miR-206*, although the level of *miR-206* declines during postnatal development in whole muscles with a high content of glycolytic fibers, since *miR-206* is most abundantly expressed in oxidative fibers (Supplementary Fig. 1a–c). *miR-1-1* and *miR-1-2* show identical mature microRNA sequences, while the related *miR-206* contains the same seed sequence critical for target binding as *miR-1-1* and *miR-1-2* and differs only by four nucleotides in the rest of the molecule[20]. Similarly, the mature nucleotide sequences of *miR-133a-1* and *miR-133a-2* are identical and differ from *miR-133b* only by one nucleotide, located outside of the seed sequence (Fig. 1a). The identical seed sequences and overlapping expression profiles of *miR-1-1*, *miR-1-2* and *miR-206* as well as of *miR-133a-1*, *miR-133a-2*, and *miR-133b* strongly suggests overlapping functions[17]. This assumption is supported by massive upregulation of *miR-206* in adult skeletal muscle after deletion of *miR-1-1/133a-2//miR-1-2/133a-1* (skeletal muscle-specific *miR-1/133a* double-knockout, dKO)[14]. Of note, loss of *miR-1/133a* caused upregulation of *miR-206* in both oxidative and glycolytic muscles, although *miR-206* is only highly expressed in oxidative fibers in wildtype muscles, further arguing for compensatory actions of *miR-206/133b* after loss of *miR-1/133a* (Fig. 1b). On the other hand, it is conceivable that the differences in a few bases in mature *miR-1* and *miR-206*, despite being outside the seed sequences, allow non-overlapping effects of these miRNAs.

To gain insight into the putative cooperative roles of *miR-1/133a* and *miR-206/133b*, we generated skeletal muscle-specific *miR-1/206/133* knockout mice (tKO) by mating *miR-206/133b* sKO to *miR-1/133a* dKO mice[14,21] (Supplementary Fig. 1d–i). No surviving *miR-1/206/133* tKO mice were observed at weaning (postnatal stage P21) (Fig. 1c, Supplementary table 1a), while germline mutants of *miR-206/133b* (*miR-206/133b* sKO) and *miR-1-1/133a-2−/−//miR-1-2/133a-1lox/lox//Pax7-Cre+/−* (*miR-1/133a* dKO) mice showed the expected Mendelian ratios and no decrease in survival rates. In contrast, *miR-1/206/133* tKO animals delivered by caesarean section at embryonic day 18.5 (E18.5) displayed the expected Mendelian distribution (Fig. 1c, Supplementary table 1b), no gross morphology abnormalities, and no differences in body weight compared to control littermates (Fig. 1d, e). Comparative transcriptome analysis of quadriceps muscles of *miR-1/133a* dKO and *miR-1/206/133* tKO in combination with gene set enrichment analysis (GSEA) revealed a highly significant enrichment of potential *miR-1/206* and *miR-133* target genes in *miR-1/133/206* tKO muscles (Fig. 1f, g). Together with the postnatal lethality of *miR-1/133/206* tKO mice, this observation suggests that the presence of *miR-206/133b* in *miR-1/133a* dKO muscle is sufficient to compensate for the absence of *miR-1/133a* and to repress a large number of *miR-1/133a* target genes. Interestingly, analysis of *miR-1/206/133* tKO mice at E18.5 revealed no signs of increased expression of the *miR-1/133a* target MEF2A, no increase of the expression of the microRNA mega-cluster at the *Dlk1-Dio3* locus, and no mitochondrial dysfunction as previously observed in adult *miR-1/*

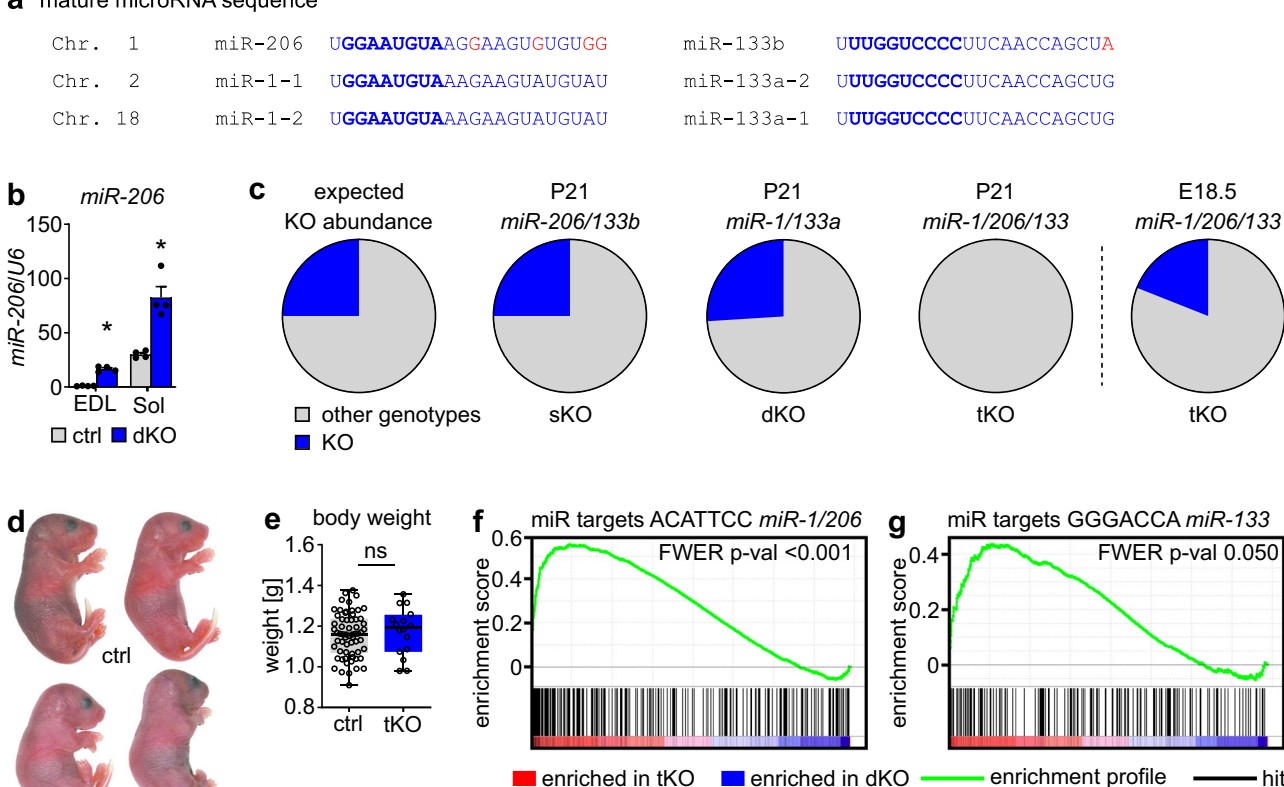

**Fig. 1 *miR-206/133b* compensate for the absence of *miR-1/133a* in skeletal muscles to prevent perinatal death. a** The miRNAs *miR-1*, *-133a/b* and *-206* are encoded in three clusters located on mouse chromosome (Chr.) 1, 2 and 18. The primary sequence of mature *miR-1/133a* is identical; *miR-206* varies from *miR-1* only by four bases, *miR-133b* differs in only one base from *miR-133a* (red). The seed sequence, essential for miRNA-target interaction, is identical in *miR-1/206* and *miR-133a/133b*, respectively (bold blue). **b** Relative expression of *miR-206* in wild-type (control) and adult *miR-1/133a* dKO *extensor digitorum longus* muscles (EDL) and *soleus* muscles (Sol) analyzed via TaqMan assay (n = 4 ctrl/4 dKO, n = 4 ctrl/4 dKO [males/females, 13–17 weeks], Mann-Whitney-U test, one-tailed, *p = 0.0143, Data are presented as mean values +/− SEM). *U6 snRNA* served as control. **c** Percentages of *miR-206/133b* sKO and *miR-1/133a* dKO animals at postnatal day 21 (P21) are in accordance to the expected Mendelian distribution (likelihood 25%), while *miR-1/206/133* tKO animals were missing at P21. Analysis of E18.5 animals reveals presence of *miR-1/206/133* tKO animals at this developmental stage (P21: n = 147 control littermates 48 sKO, n = 100 control littermates/35 dKO, n = 34 control littermates/0 tKO at P21; E18.5: n = 26 control littermates 6 tKO). **d** Gross morphology of control and *miR-1/206/133* tKO pups after caesarean section (control = littermates, scale-bar: 1 cm [E18.5]). **e** Box-plot of *miR-1/206/133* tKO and control body weight (n = 63 control/16 tKO, control = littermates, Mann-Whitney-U test, one-tailed, ns = not significant, Box indicates median and 25th to 75th percentiles, whiskers indicate minimum and maximum [E18.5]). **f, g** GSEA of *miR-1/133/206* tKO compared to *miR-1/133a* dKO quadriceps muscle transcriptome data reveals enrichment of predicted *miR-1/206* and *miR-133* targets in *miR-1/133/206* tKO (n = 4 dKO/4 tKO, familywise-error rate method (FWER), p = <0.001 (**f**), p = 0.05 (**g**) [E18.5]). Source data are provided as a Source Data file.

*133* dKO mice[14] (Suppl. Fig. 2). We concluded that the function of *miR-1/133a* strongly depends on the developmental stage and that the repression of MEF2A *miR-1/133a* is limited to adult skeletal muscles.

**miR-1/206/133 tKO impairs NMJs but not muscle development**. Most *miR-1/133/206* tKO mutants from heterozygous mutant parents at E18.5 showed only weak movements upon physical stimulation. Furthermore, *miR-1/133/206* tKO mutant lungs mutants failed to float 15 min after caesarean section (Fig. 2a, b), indicating respiratory failure, probably due to the inability of skeletal muscles to expand the lungs. Surprisingly, however, histological analysis did not reveal any obvious changes in skeletal muscle morphology in *miR-1/133/206* tKO mutants at E18.5 (Supplementary Fig. 3a–d). Likewise, the average fiber number and the fiber size distribution were normal in both primarily glycolytic and oxidative muscles (Supplementary Fig. 3e–j) and no substantial alterations in the expression of markers for myogenic development and differentiation were detected in *miR-*

*1/206/133* tKO mice (Supplementary Fig. 3k). Ultrastructural analysis by transmission electron microscopy showed well-developed sarcomere structures with clearly visible I-bands and H-zones in skeletal muscles of *miR-1/133/206* tKO mutants that did not deviate from controls (Supplementary Fig. 3l). To investigate potential defects during embryogenesis that might be compensated at later stages of skeletal muscle development, we analyzed *miR-1/206/133* deficient animals at E10.5. We detected no differences in the expression and spatial distribution of Myogenin (*Myog*), a marker of early myogenic differentiation, between *miR-1/206/133* and control animals at E10.5. Identical results were obtained with *miR-1/206/133* germline knock-out animals (gtKO; obtained after mating of *miR-1/133a* constitutive KO[12] to *miR-206/133b* sKO), which were still viable at this stage (Supplementary Fig. 4a).

Next, we tested the ability of primary *miR-1/206/133* tKO muscle stem cells (MuSC) to differentiate in vitro after tamoxifen-induced deletion of *miR-1-2/133a-1* in adult *miR-206/133b⁻/⁻//miR-1-1/133a-2⁻/⁻//miR-1-2/133a-1^lox/lox//Pax7-CreERT2⁺/⁻* mice (MuSC tKO, Supplementary Fig. 4b–k). *miR-*

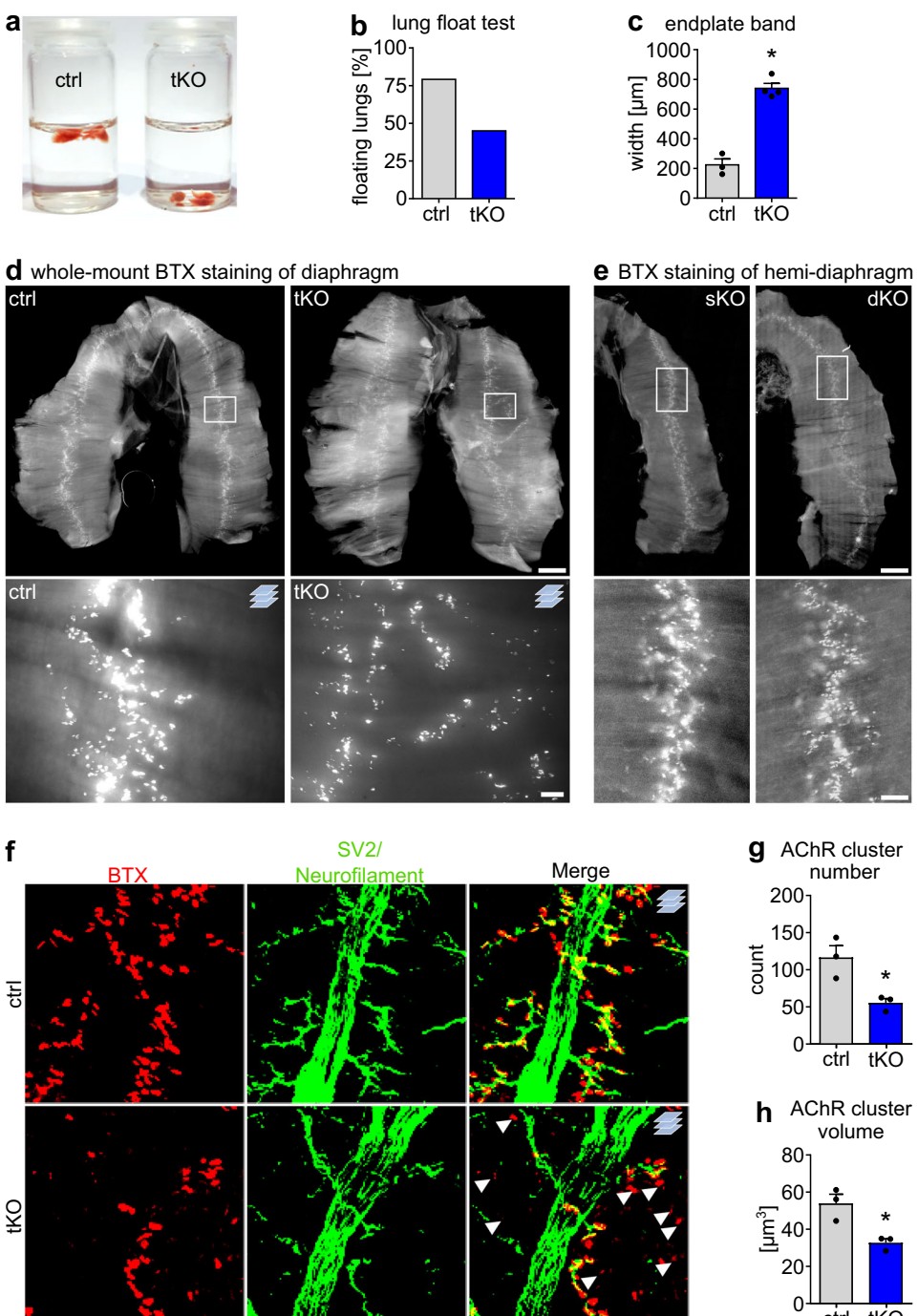

**Fig. 2 Loss of *miR-1/206/133* in skeletal muscles disturbs neuromuscular synapse formation and causes respiratory failure. a** Lung floating test of control and *miR-1/206/133* tKO reveals deficits in breathing after recovery by caesarean section in tKO animals compared to control littermates [E18.5]. **b** Representation of percentages of control and *miR-1/206/133* tKO in the lung floating test (percent of floating lungs in each group; n = 28 control/11 tKO from 8 litters, control = littermates [E18.5]). **c** Measurement of average endplate band width of control and *miR-1/206/133* tKO diaphragms (n = 3 control/ 4 tKO, control = littermates, Mann-Whitney-U test, one-tailed, *p = 0.0286, [E18.5]). **d** Whole-mount bungarotoxin (BTX) staining of diaphragms from control and *miR-1/206/133* tKO animals (control = littermates, scale-bar: 600 µm upper row/50 µm lower row [50 µm thick Z-stacks lower row, n = 3 control/4 tKO, [E18.5]). **e** Whole mount BTX staining of hemi-diaphragms of *miR-206/133b* sKO (n = 3) and *miR-1/133a* dKO (n = 2) animals (scale-bar: 600 µm upper row/100 µm lower row [E18.5]). **f** Z-stacks of a whole mount-stained diaphragms from control (littermate) and *miR-1/206/133* tKO mice. Anti-SV2 (synaptic vesicles) and anti-neurofilament antibodies were used to label both nerve terminals and axons (green). Staining of the postsynapse was done using BTX (red). Arrows indicate AChR clusters without a nerve terminal in close proximity (scale-bar: 20 µm, n = 3 control/3 tKO [E18.5]). **g**, **h** Quantification of the average AChR cluster number and AChR volume in BTX-stained whole mount diaphragms based on Z-stacks, comparing *miR-1/206/133* tKO and control samples (Supplementary Fig. 5b). Data were obtained from 3D rendered confocal Z-stacks (n = 3 control/3 tKO, control = littermates, Mann-Whitney-U test, one-tailed, *p = 0.05, [E18.5]). Source data are provided as a Source Data file. Data are presented as mean values +/− SEM in **c** and **g**, **h**.

1/206/133 tKO and control (*miR-1-1/133a-2⁻/⁻//miR-1-2/133a-1^{lox/lox}//miR-206/133b⁻/⁻//Pax7-CreERT2^{+/+}*) MuSCs derived from mice with tamoxifen treatment showed the same proliferation rate over 120 h in culture and displayed neither differences in myotube formation nor altered expression of important myogenic factors and other markers of differentiation (Supplementary Fig. 4l–n). Taken together, these results exclude major functions of *miR-1/206/133* for skeletal muscle development, MuSC formation and differentiation.

Since neonatal *miR-1/206/133* tKO mice exhibited normally developed skeletal musculature but only weak movements and eventually died, we wondered whether innervation defects are responsible for the failure to expand the lungs. Therefore, we analyzed innervation of the diaphragm. Whole-mount staining of the diaphragm muscle with Alexa Fluor555-conjugated α-Bungarotoxin (BTX) revealed striking defects in the formation of postsynaptic NMJs in *miR-1/206/133* tKO mice as indicated by dispersed localization of AChR signals (Fig. 2c, d). Furthermore, we detected massive changes in the presynapsis of *miR-1/206/133* tKO diaphragms. Axons were not restricted to a narrow endplate band, but extended to a broader region (Supplementary Fig. 5a). In contrast, *miR-206/133b* sKO and *miR-1/133a* dKO diaphragms displayed a normal pattern of innervation and no alterations of NMJ formation (Fig. 2e). Quantitative analysis of AChR-clusters showed a strong reduction of BTX-stained NMJ endplates in *miR-1/206/133* tKO diaphragm and paraspinal muscles (Fig. 2f, Supplementary Fig. 5b–d). We observed not only a decline of the AChR cluster number (Fig. 2g), but also a significant reduction of the AChR cluster size in both diaphragm and paraspinal muscles (Fig. 2h, Supplementary Fig. 5e–f). In agreement with the morphological assessment, transcriptome data of synaptic marker genes indicated an upregulation of key postsynaptic molecules (Supplementary Fig. 5g) and GSEA unveiled a highly significant enrichment of gene sets associated with neuromuscular synapse formation and assembly in *miR-1/206/133* tKO quadriceps muscles (Fig. 3a–c). We reasoned that the increased expression of synaptic genes represents a secondary effect due to reduced innervation of tKO muscle[22,23]. We concluded that the loss of all *miR-1/206/133* clusters impairs formation of NMJs but otherwise has no discernible effects on myofiber differentiation.

**miR-1/206/133 adjust the concentration of CRK at NMJs.** To identify primary targets of *miR-1/206/133* and thereby gain insight into the molecular mechanism by which these miRNAs control NMJ formation, we analyzed the overlap of *miR-1/206/133* tKO transcriptome data, predicted *miR-1/206/133* target genes, and known NMJ-associated genes. *Crk* emerged out of this analysis as a potentially relevant primary *miR-1/206/133* target (Fig. 3d, Supplementary table 2). CRK is an adapter protein that is ubiquitously expressed and fulfills important roles in several signaling pathways by mediating signaling events downstream of receptor tyrosine kinases[24–26]. In skeletal muscle, CRK and CRK-L play a critical role during formation of neuromuscular synapses downstream of the MuSK-binding protein DOK7[4,7,8] Interestingly, CRK was specifically enriched at NMJs in myofibers (Fig. 3e)[4] and even more importantly, inactivation of *miR-1/206/133* caused a profound increase of CRK mRNA and protein levels in skeletal muscles (Fig. 3f–h). Inspection of the nucleotide sequence of CRK uncovered target sites for *miR-1/206* and *miR-133* in the 3′-UTR of murine *Crk* mRNA, which were highly conserved in human *Crk* (Fig. 3i). To confirm the putative repression of *Crk* by *miR-1/206/133*, we studied the functional relevance of the most strongly conserved *miR-1/206* and *miR-133* target sites using luciferase reporter assays in C2 myoblasts and myotubes (Fig. 3j, k). After transfection of the luciferase reporter

constructs, cell lysates were prepared from either proliferating myoblasts or differentiating myotubes, since myoblasts express much lower amounts of *miR-1/133* than differentiated myotubes (Fig. 3l, m)[14,19,27]. In line with the potential repression of *Crk* by *miR-1/206/133*, activity of the luciferase reporter was much lower in proliferating than in differentiated myotubes. The reduction of luciferase reporter activity in myoblast compared to myotubes was not apparent anymore when the predicted miRNAs target sites were mutated (Fig. 3n, o), suggesting that endogenous levels of *miR-1/206/133* mediate repression of *Crk* in myotubes.

**High CRK activates RAC1 via FARP1, disturbing NMJ function.** Previous studies reported a role of CRK downstream of LRP4/MuSK/DOK7, probably by recruitment of GEFs (guanine nucleotide exchange factors) to NMJs[4,28], which might be instrumental for activation of small GTPases such as RAC and RHO. Since these small GTPases are essential for formation of agrin-induced AChR-micro- and macro-clusters in vitro[29,30], we wondered whether CRK is the missing link that connects MuSK with RAC or RHO activation[9]. Immunoprecipitation of CRK followed by mass spectrometry revealed an interaction of CRK with the RAC1-GEF FARP1[31]. The amount of co-immunoprecipitated FARP1 was higher in E18.5 *miR-1/206/133* tKO limb muscles compared to WT (Suppl. Fig. 6a–c), suggesting that loss of *miR-1/206/133* increases CRK-FARP1 interactions leading to enhanced RAC1 activity. To confirm enhanced interaction of CRK and FARP1 in *miR-1/206/133* tKO muscle, we performed proximity ligation assays (PLA) using myofibers differentiated from control and *miR-1/206/133* deficient MuSCs (MuSC *miR-1/206/133* tKO). The PLA assay detected increased amounts of endogenous CRK in close proximity to the RAC1-GEF FARP1 in differentiated *miR-1/206/133 tKO* muscle fibers (Supplementary Fig. 6d–f). The interaction of CRK with FARP1 was validated by co-immunoprecipitation using a FARP1 antibody again (Supplemntary Fig. 6g). In addition, mass spectrometry-based proteomics of control and *miR-1/206/133* tKO quadriceps muscles revealed enrichment of RAC GTPase binding proteins in *miR-1/206/133* tKO muscles (Fig. 4a, Supplementary Fig. 6h, i). We also observed enhanced RAC-GTP downstream signaling at the transcriptional level via GSEA in *miR-1/206/133* tKO muscles (Supplementary Fig. 7a, b), which corroborates that loss of *miR-1/206/133* and the subsequent rise in CRK increases activity of small GTPases in *miR-1/206/133* tKO muscles. While expression levels of *miR-1/206/133* have an instructive role for AChR-clustering, agrin-induced AChR clustering does not affect expression of *miR-1/206/133* (Supplementary Fig. 8a–c).

To explore in more detail the impact of increased CRK levels on RAC1 GTPase activity, we performed pull down assays for active RAC1 (RAC1-GTP). Forced expression of CRK in C2 cells strongly activated RAC1 in myotubes (Fig. 4b) and impaired formation of agrin-induced AChR-macro-clusters (Fig. 4c, d), indicating that the amount of CRK needs to be tightly regulated for achieving the appropriate degree of RAC1 activation. In contrast, increased expression of the CRK paralog CRK-L did not affect AChR formation, showing the same lack of effects as control transfections (Supplementary Fig. 8d–g). Similar to *Crk* transfections, expression of a constitutive active version of RAC1 (RAC^{G12V}) also impaired AChR-macro-cluster formation (Fig. 4e, f)[30], further demonstrating that increased levels of CRK enhance recruitment of FARP1 to forming AChR-macro-clusters leading to overactivation of RAC1, which impairs AChR-macro-cluster formation in vitro.

**Forced *Crk* expression disrupts NMJ formation in vivo.** To explore whether loss of *miR-1/206/133*-mediated repression of CRK is sufficient to disturb formation of NMJ, we overexpressed CRK in

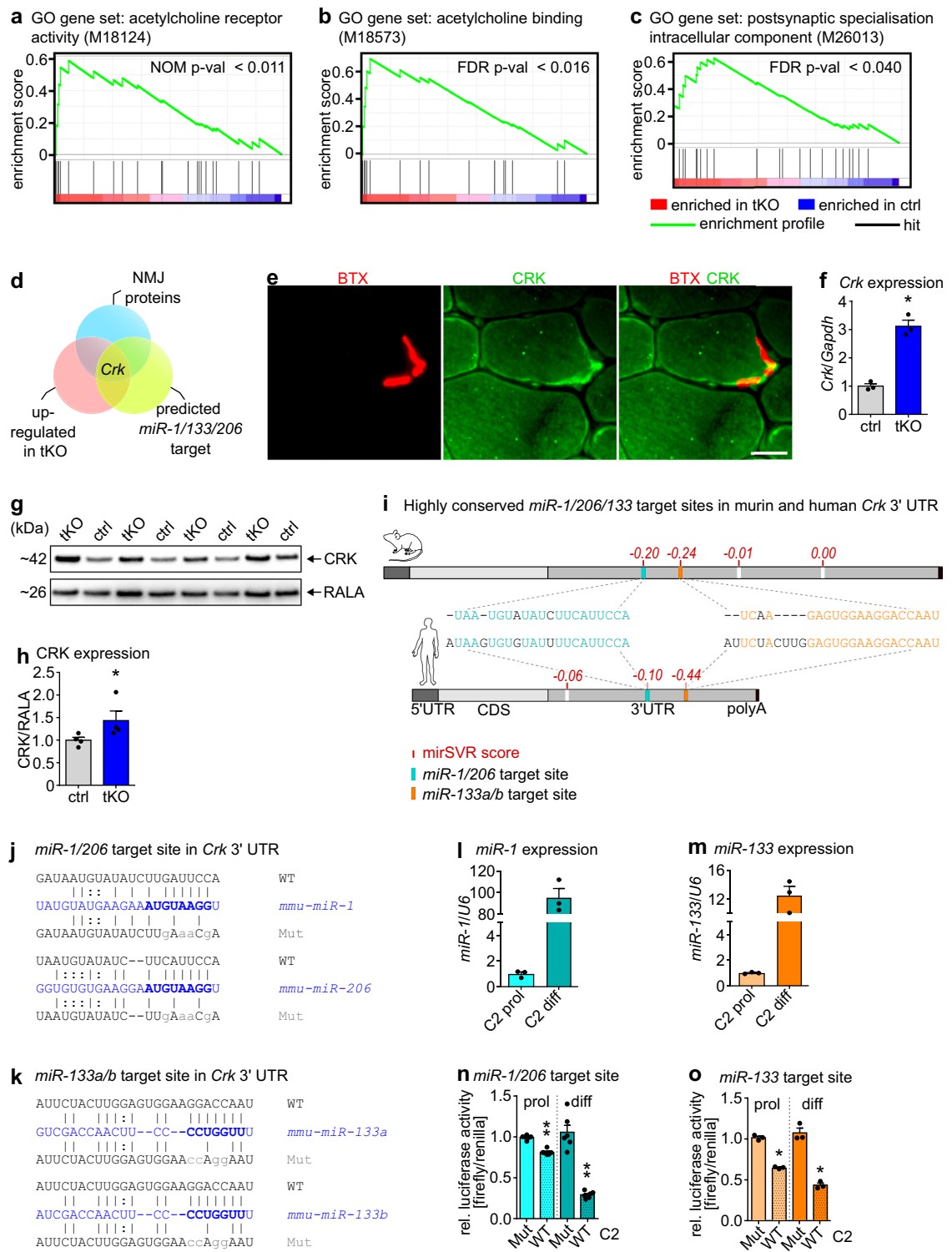

**a** GO gene set: acetylcholine receptor activity (M18124) — NOM p-val < 0.011

**b** GO gene set: acetylcholine binding (M18573) — FDR p-val < 0.016

**c** GO gene set: postsynaptic specialisation intracellular component (M26013) — FDR p-val < 0.040

enriched in tKO | enriched in ctrl | enrichment profile | hit

**d** NMJ proteins / up-regulated in tKO / predicted miR-1/133/206 target — Crk

**e** BTX | CRK | BTX CRK

**f** Crk expression — Crk/Gapdh — ctrl, tKO *

**g** (kDa) tKO ctrl tKO ctrl tKO ctrl tKO ctrl ← CRK ~42, ← RALA ~26

**h** CRK expression — CRK/RALA — ctrl, tKO *

**i** Highly conserved miR-1/206/133 target sites in murin and human Crk 3' UTR

**j** miR-1/206 target site in Crk 3' UTR

**k** miR-133a/b target site in Crk 3' UTR

**l** miR-1 expression — miR-1/U6 — C2 prol, C2 diff

**m** miR-133 expression — miR-133/U6 — C2 prol, C2 diff

**n** miR-1/206 target site — rel. luciferase activity [firefly/renilla]

**o** miR-133 target site — rel. luciferase activity [firefly/renilla]

skeletal muscles in vivo in transgenic mice. Due to the anticipated lethality of CRK overexpression, we recovered 61 animals at E18.5, which were injected at the pronuclear stage with a CRK-construct under control of a skeletal muscle-specific promoter[32]. The 3′-UTR containing the miR-1/206/133 target sequences was not included to avoid counter-regulation by miR-1/206/133. 16% of injected animals carried the transgene (Crk-tg) (Fig. 5a, Supplementary Fig. 9a) and expressed varying concentrations of CRK in skeletal muscles as measured by western blot analysis (Fig. 5b, c). We also created a second mouse model of Crk overexpression, based on the CRISPR-

dCas9-SPH system[33], to enable temporally controlled over-expression of Crk in embryonic and adult skeletal muscles after activation by skeletal muscle-specific expression of Cre-recombinase. sgRNAs targeting the transcriptional start site of Crk[34] were tested in C2 myoblasts using the dCAS9-VPH system[35]. The most effective sgRNA (Fig. 5d) was selected to generate U6-sgRNA transgenic mice. U6-sgRNA$^{+/-}$//Pax7-ICN-Cre$^{+/-}$//CAG-LSL-dCas9-SPH$^{+/-}$ (Crk-SPH) mice (Fig. 5e, Suppl. Fig. 9b–d), showing dCas9-SPH activity and massively increased Crk expression in skeletal muscles (Fig. 5f, g), were recovered at E18.5. Importantly, we observed

**Fig. 3 The NMJ associated adapter protein _Crk_ contains evolutionary conserved miR-target sites in the 3´UTR and is a target for _miR-1/206/133_.** **a–c** GSEA of control and _miR-1/206/133_ tKO quadriceps muscle transcriptome data ($n = 4$ control/4 tKO, control = _miR-1-1/133a-2$^{+/+}$//miR-1-2/133a-1$^{lox/}$_ $^{lox}$//miR-206/133b$^{+/+}$//Pax7-Cre$^{+/+}$, familywise-error-rate method (FWER), $p = <0.011$ (**a**), $p = <0.016$ (**b**), $p = <0.040$ (**c**) [E18.5]). GO = Gene Ontology. **d** Overlap of transcriptome data (significantly up-regulated in tKO with fold change >1.35) and predicted _miR-1/206/133_ target genes identified the NMJ-associated gene _Crk_ as a top candidate. **e** CRK immunostaining reveals localization at the postsynapse in transversal sections of WT TA-muscles (scale-bar: 20 µm, BTX = bungarotoxin, $n = 2$ [male, 14 weeks]). **f** _Crk_ expression in _miR-1/206/133_ tKO deficient muscles compared to control quadriceps (TaqMan assay, $n = 3$ control/3 tKO, control= _miR-1-1/133a-2$^{+/+}$//miR-1-2/133a-1$^{lox/lox}$//miR-206/133b$^{+/+}$//Pax7 Cre$^{+/+}$_, Mann-Whitney-U test, one-tailed, *$p = 0.05$, [E18.5]). _Gapdh_ served as control. **g, h** Western blot analysis of CRK expression in _miR-1/206/133_ deficient diaphragm muscles (tKO) compared to control. RALA served as control. ($n = 4$ control/4 tKO, control= _miR-1-1/133a-2$^{+/+}$//miR-1-2/133a-1$^{lox/lox}$//miR-206/133b$^{+/+}$//Pax7-Cre$^{+/+}$_, Mann-Whitney-U test, one-tailed, *$p = 0.0143$, [E18.5]). **i** Conserved predicted binding sites in murine and human _Crk_ mRNA. Putative _miR-1/206_ target sites (turquoise) and _miR-133_ binding sites (orange) in the 3´UTR. MicroRNA target sites ranked by mirSVR down-regulation score (red). CDS = coding sequence, UTR = untranslated region, polyA= polyadenylation-tail. **j, k** WT and mutated (Mut) sequences (gray) of putative _miR-1/206_ (**j**) and _miR-133a/b_ (**k**) target sites in the 3´UTR of _Crk_ were cloned into luciferase reporter vectors. MiR-sequence (blue), seed-sequence (bold blue). **l, m** Endogenous _miR-1_ or _miR-133_ expression in proliferating (prol) and differentiation stages (diff) of C2 cells (TaqMan assays, $n = 3$ prol/3 diff, $n = 3$ prol/3 diff, biological replicates). _U6_ served as control. **n, o** Repression of firefly-luciferase activity in Dual-Luciferase reporter assays by WT but not mutated (Mut) _miR-1_ or _miR-133_ miRNA target sites from 3´UTR of _Crk_. Repression of firefly-luciferase by the respective WT target sites is detected in proliferating, but is more pronounced in differentiating (diff) C2 cells (_miR-1/206_ target site $n = 6$ Mut/6 WT, _miR-133_ target site $n = 3$ Mut/3 WT, biological replicate = independent transfection, Mann-Whitney-U test, one-tailed, **$p = 0.0011$, *$p = 0.05$). Normalization via renilla-luciferase (endogenous control). Source data are provided as a Source Data file. Data are presented as mean values $+/-$SEM in **f–h** and **l–o**.

disturbed distribution of BTX-positive postsynaptic AChR-clusters in the diaphragm of both the _Crk-tg_ and the _Crk-SPH_ models at E18.5, very similar to the _miR-1/206/133_ tKO phenotype. Not only the distribution of AChR-clusters was much more dispersed (Fig. 5h), but also the AChR cluster number was strongly reduced (Fig. 5i). Consistent with the severe impairment of postsynaptic development, _Crk-SPH_ mice did not survive early postnatal development but similar to _miR-1/206/133_ tKO animals presented with normal body weight and normal gross morphology at E18.5 (Fig. 5j–l; Supplentary Table 3). GSEA of transcriptome data of _Crk_-overexpressing animals compared to littermates revealed enrichment of gene sets associated with synapse assembly and function (Fig. 5m, n). Likewise, expression of FARP1 was increased in _Crk-SPH_ mice (Supplementary Fig. 6i), further supporting the relevance of the CRK-FARP1-RAC1 in formation of NMJs. Taken together, recapitulation of the _miR-1/206/133_ tKO phenotype in two independent models of increased _Crk_ expression strongly suggests that tight control of the adaptor protein CRK is a crucial function of _miR-1/206/133_ during early postnatal skeletal muscle development to achieve proper NMJ formation.

**Maintenance of NMJ depends on _miR-1/206/133_ in adult muscles.** Our finding that _miR-1/206/133_ tKO mice at E18.5 do not show increased expression of the microRNA mega-cluster at the _Dlk1-Dio3_ locus and mitochondrial dysfunction as adult _miR-1/133_ dKO mice[14], strongly suggest that the function of _miR-1/133a_ depends on the developmental stage. Thus, we decided to interrogate the potential redundancy of _miR-206/133b_ with _miR-1/133a_ miRNAs and the role of _miR-1/206/133_ for NMJ maintenance in adult skeletal muscles. To accomplish efficient deletion of _miR-1/206/133_ in adult skeletal myofibers, we used _HSA-rtTA/TRE-Cre_ mice[36] in combination with the _miR-206/133b$^{-/-}$//miR-1-1/133a-2$^{-/-}$//miR-1-2/133a-1$^{lox/lox}$_ strain (adult _miR-1/206/133_ tKO). Efficiency of Cre-recombination was verified by doxycycline treatment of _HSA-rtTA/TRE-Cre$^{+/-}$//Z/AP-Cre$^{+/-}$_ reporter mice for three weeks (Supplementary Fig. 10a)[37], which was sufficient to achieve complete recombination of the reporter construct in skeletal muscle fibers (Supplementary Fig. 10b). TaqMan analysis confirmed a strong reduction of _miR-1/133a_ expression in TA and soleus muscles of adult _miR-1/206/133_ tKO mice nine weeks after starting the three weeks doxycycline treatment (Fig. 6a–e). Furthermore, we repeated the doxycycline treatment regimens every six weeks to ensure that new nuclei contributed by fusing myoblasts will also undergo recombination. Expression analysis by Affymetrix

microarray and TaqMan analyses of adult _miR-1/206/133_ tKO skeletal muscles revealed a strong enrichment of _miR-1_ and _miR-133_ target gene sets in skeletal muscle 28 weeks after initiation of doxycycline administration (Fig. 6f, g), including upregulation of _Crk_ (Fig. 6h). Consistent with observations in neonatal _miR-1/206/133_ tKO mice, we also observed activation of RAC1 downstream signaling in adult _miR-1/206/133_ tKO animals by transcriptional profiling and GSEA (Supplementary Fig. 11a, b). To directly compare adult _miR-1/206/133_ tKO animals to mice expressing increased levels of CRK in myofibers, we generated adult _Crk-SPH_ mice (_U6-sgRNA$^{+/-}$//HSA-rtTA/TRE-Cre$^{+/-}$//CAG-LSL-dCas9-SPH$^{+/-}$_) allowing doxycycline-dependent control of _Crk_ expression (Fig. 6i). Western blot analysis confirmed strongly increased CRK levels in skeletal muscles of adult _Crk-SPH_ mice, one week after completion of the 3-weeks long doxycycline treatment (Fig. 6j). Overexpression of CRK in adult skeletal muscles led to activation of RAC1 downstream targets, such as PAK1/2/3, corroborating elevated RAC1 activity in response to increased CRK expression (Supplementary Fig. 11c–e).

Importantly, both adult _miR-1/206/133_ tKO and adult _Crk-SPH_ mice displayed severe and progressive deficits in motor function, which in case of adult _Crk-SPH_ mice became already apparent one week after completion of the doxycycline treatment. We observed muscle weakness, abnormal hindlimb clasping after lifting mice at the tail, and massive disturbance of limb coordination as indicated by footprint assays (Fig. 7a)[38–40]. Similarly, adult _miR-1/206/133_ tKO and adult _Crk-SPH_ mice showed reduced performance in Rotarod tests (Fig. 7b–e)[38,41]. In line with these findings, molecular analysis revealed strong enrichment of gene sets involved in organization of the postsynapse in skeletal muscles of adult _miR-1/206/133_ tKO mice (Fig. 7f, g). Of note, deficits in motor function were not apparent in adult _miR-1/133a_ dKO and _miR-206/133b_ KO mice (Supplementary Fig. 12a–c, Fig. 7a), while adult _miR-1/206/133_ tKO showed the same upregulation of microRNA mega-cluster at the _Dlk1-Dio3_ locus and mitochondrial dysfunction as adult _miR-1/133a_ dKO (Supplementary Fig. 13). Impaired neuromuscular functions corresponded well to morphological changes of postsynaptic NMJ regions in skeletal muscles. We detected a significant decrease of the number of BTX-stained postsynaptic NMJ regions in adult _miR-1/206/133_ tKOs and adult _Crk-SPH_ animals. Furthermore, we observed fragmentation of pretzel-shaped branches of AChR clusters in isolated single fibers as well as in whole muscle preparations (Fig. 7h–j, Supplementary Fig. 14). Taken together, these results indicate that adjustment of CRK-levels

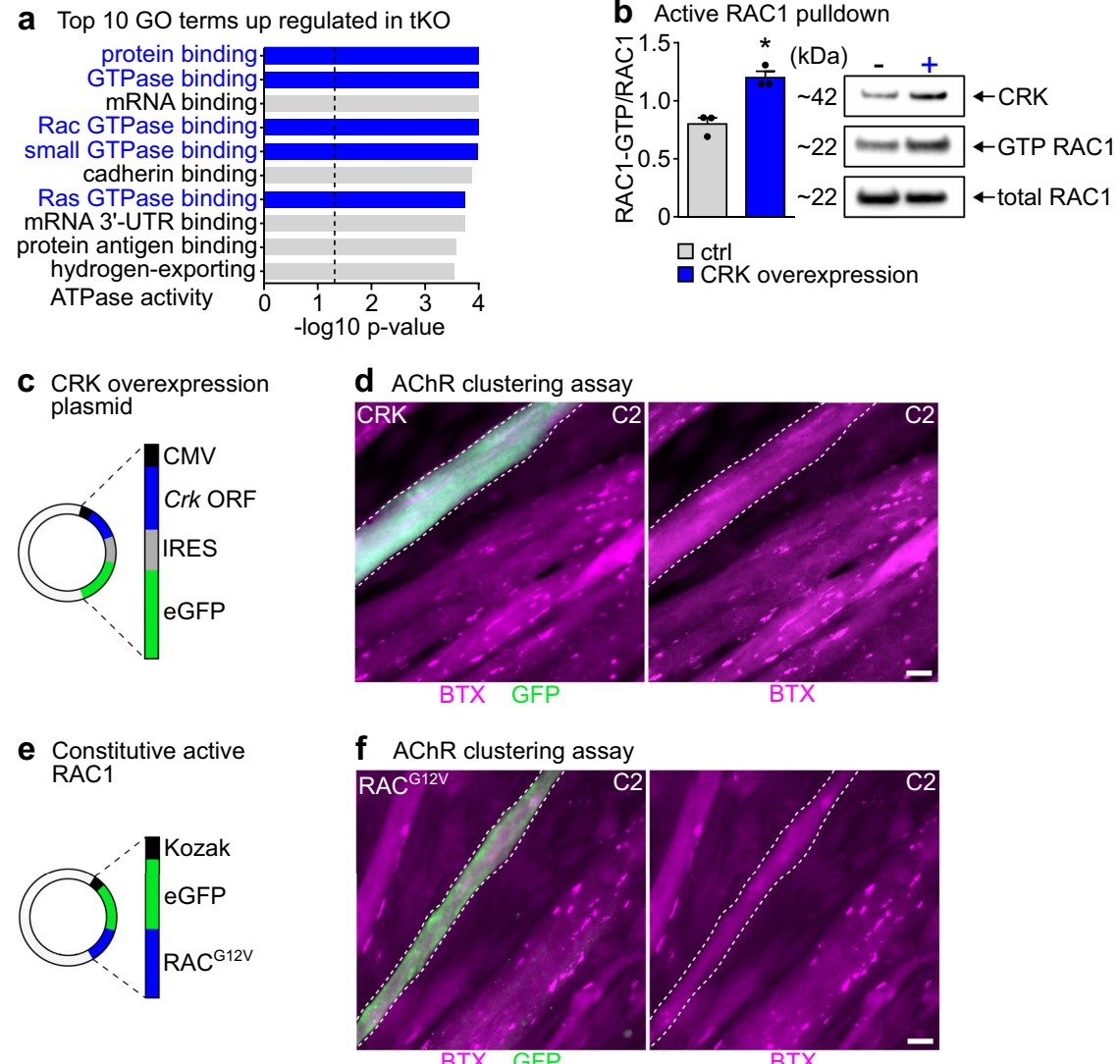

**Fig. 4 Increased levels of CRK in skeletal muscle cells activate RAC1, resulting in impaired AChR clustering. a** Top 10 up regulated GO (Gene Ontology) terms of molecular function in Proteome data indicate increased small GTPase activity (terms labelled in blue) in *miR-1/206/133* tKO vs. WT proteome obtained from Quadriceps muscles at E18.5 (*n* = 3 WT/3 tKO). **b** Active RAC1 pulldown assays exhibit significant increase of RAC1 activation (RAC1-GTP) upon CRK overexpression in skeletal muscle cells (*n* = 3/3 independent transfections and pulldown assays, loaded on three gels, Mann-Whitney-U test, one-tailed, *\*p* = 0.05, Data are presented as mean values +/− SEM). RAC1-GTP levels were normalized to total RAC1 levels using Western blot analysis. **c, d** *Crk* was ectopically expressed in C2 myotubes using a vector with CMV promoter and IRESII-mediated GFP co-expression (**c**). Transfected myotubes were incubated with agrin to induce AChR clustering. GFP-positive (anti-GFP-Alexa488), *Crk*-overexpressing myotubes loose BTX-positive AChR-macro-clusters (stained by Alexa Fluor594-conjugated BTX; scale-bar: 20 μm). Dashed lines mark myotubes expressing *Crk-GFP* (**c**) and *RAC^{G12V}-GFP* (**d**). Minimum *n* = 6 (biological replicate = independent transfection). **e, f** Constitutive active RAC1 (RAC^{G12V}) was ectopically expressed in C2 myotubes. Transfected myotubes were treated with agrin to induce AChR clustering. RAC^{G12V}-GFP (Alexa488) positive myotubes loose AChR-macro-clusters (stained by Alexa Fluor594-conjugated BTX; scale-bar: 20 μm). Minimum *n* = 6 (biological replicate = independent transfection). Source data are provided as a Source Data file.

by *miR-1/206/133* is not only essential for NMJ formation but also critical for maintaining neuromuscular functions in adult mice, probably by controlling RAC1 activity via limiting recruitment of FARP1 to NMJs (Fig. 8). Surprisingly, this pivotal mechanism, necessary for enabling locomotion, depends on the combined activity of three *miR-1/206/133* gene clusters.

## Discussion

Here we demonstrate that CRK levels in skeletal muscle myofibers need adjustment by *miR-1/206/133*-mediated posttranscriptional regulation to establish the necessary degree of localized RAC1 activity required for NMJ formation and maintenance. Our study

reveals that *miR-1/206/133* are instrumental to customize expression of the ubiquitous adapter protein CRK to the needs of myofibers, thereby enabling proper AChR clustering. Furthermore, we show that CRK is the missing link that connects the LRP4-MuSK-DOK7 signaling complex to localized RAC1 activity at newly forming AChR-clusters by recruitment of GEF FARP1, eventually enabling formation of functional NMJs. This signaling mechanism is not only critical for postsynaptic NMJ development but also indispensable for maintenance of NMJs in adult skeletal muscles.

CRK is a universal adapter protein, essential for several signaling pathways in different cell types[42,43], which also serves specific functions for NMJ formation in skeletal muscles together

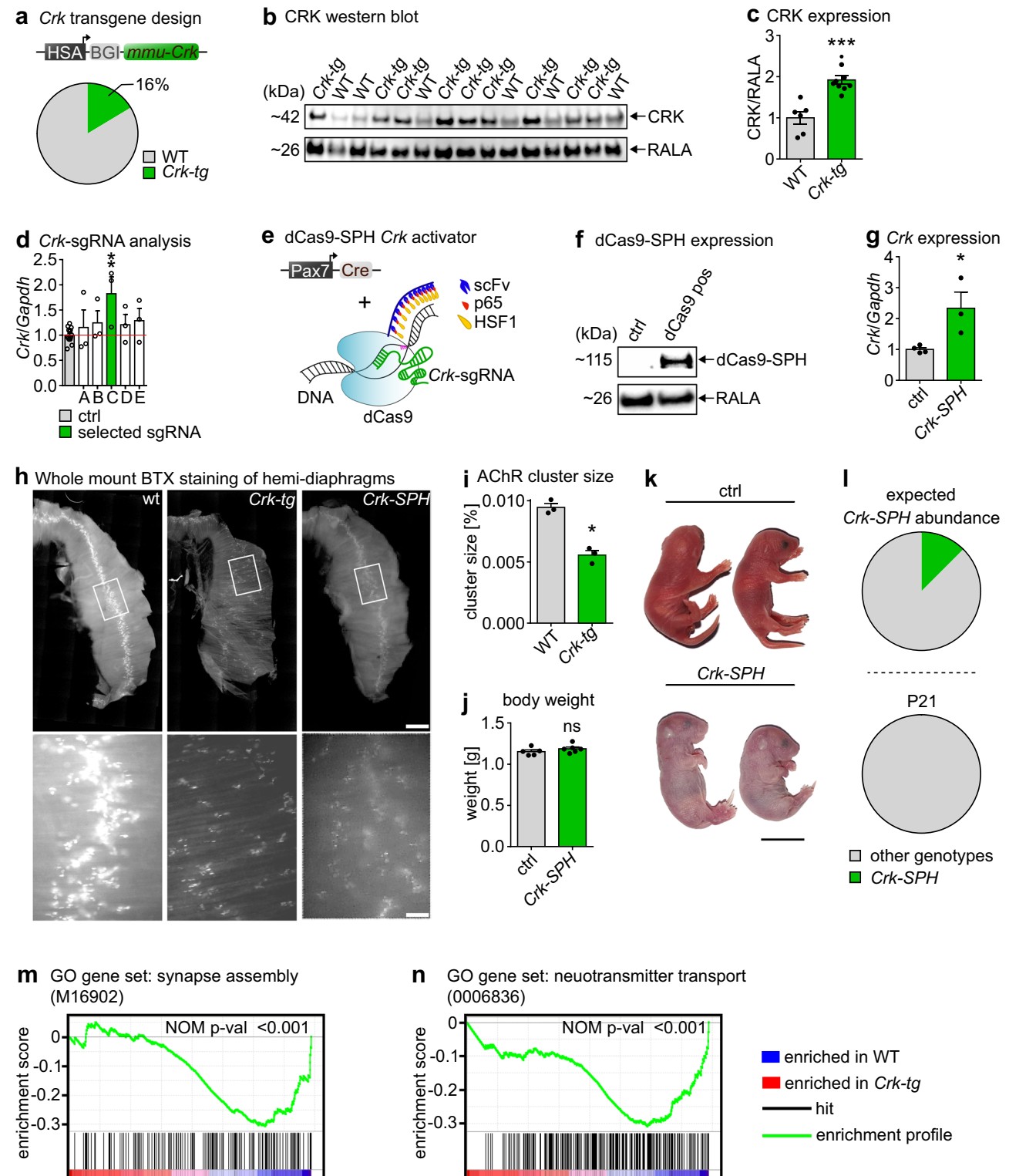

**a** *Crk* transgene design

**b** CRK western blot

**c** CRK expression

**d** *Crk*-sgRNA analysis

**e** dCas9-SPH *Crk* activator

**f** dCas9-SPH expression

**g** *Crk* expression

**h** Whole mount BTX staining of hemi-diaphragms

**i** AChR cluster size

**j** body weight

**k** ctrl / *Crk-SPH*

**l** expected *Crk-SPH* abundance

**m** GO gene set: synapse assembly (M16902)

**n** GO gene set: neuotransmitter transport (0006836)

with its paralog CRK-L[4]. Previous work based on loss-of-function approaches revealed that NMJ formation requires the presence of CRK and CRK-L, but did not appreciate the necessity to adjust CRK levels in a relatively narrow range. We found that post-transcriptional repression of CRK by *miR-1/206/133* is necessary to avoid detrimental effects on NMJ formation caused by increased levels of CRK. It was known that CRK acts downstream of the LRP4/MuSK/DOK7 signaling complex, but effects mediated by CRK were unclear[2]. We found that CRK interacts with

FARP1 probably via its SH3 (Src homology 3) domain, which acts as a recognition site for proline-rich motifs of several interaction partners[26,44]. In addition, CRK contains a SH2 (Src homology 2) domain that allows binding to DOK7[4,7,45]. CRK was recently reported to interact not only with DOK7 but also directly with the tyrosine-phosphorylated JM region of MuSK. This redundancy of CRK recruitment highlights the importance of balanced CRK levels at NMJs[45]. FARP1 is known as a GEF, stimulating RAC1 activity in neuronal postsynaptic spines downstream of SynCAM

**Fig. 5 Forced expression of *Crk* in skeletal muscles recapitulates the *miR-1/206/133* knock-out phenotype. a** Muscle-specific *Crk* overexpression by the HSA-promoter. 16% of all obtained animals were transgene-positive tested (*n* = 51 WT/10 *Crk-tg* [E18.5]). **b**, **c** Increased CRK levels in *Crk-tg* quadriceps muscles. RALA served as loading control (Western blot, n = 6 WT/9 *Crk-tg*, WT = littermates, Mann-Whitney-U test, two-tailed, ***p = 0.0004 [E18.5]). **d** Co-transfection of SP-dCas9-VPR64 vector and sgRNA vectors for *Crk* activation in C2 cells. Selected sgRNA C (green) for further analysis (TaqMan assay, control = transfections of unrelated sgRNAs *n* = 15, sgRNAs A-E tested in independent transfections *n* = 3, Mann-Whitney-U test, one-tailed, **p = 0.0049). *Gapdh* served as control. **e** *Pax7-Cre*-driven dCas9-SPH[33] activation system with selected *Crk*-sgRNA. **f** *Pax7-Cre*-dependent dCas9-SPH expression in quadriceps muscles of *Pax7-Cre*$^{+/-}$//*CAG-LSL-dCas9-SPH*$^{+/-}$ mice (Western blot, *n* = 3 control/3 *Crk-SPH*, control = sgRNA$^{+/+}$ [E18.5]). RALA served as control. **g** Increased expression of *Crk* in quadriceps muscle using the *Pax7-Cre*-dependent *dCas9-SPH* system in combination with *Crk*-sgRNA expression compared to control (TaqMan assay, *n* = 4 control/3 *Crk-SPH*, control = sgRNA$^{+/+}$, Mann-Whitney-U test, one-tailed, *p = 0.0286 [E18.5]). *Gapdh* served as control. **h** Whole-mount bungarotoxin (BTX) staining of hemi-diaphragms of *Crk* overexpressing *Crk-tg* (*n* = 9) and *Crk-SPH* (*n* = 3) animals compared to WT (scale-bar: 600 μm/100 μm [E18.5]). **i** Mean AChR cluster size in paraspinal muscles on transversal sections of WT (*n* = 3) and *Crk-tg* (*n* = 3) animals (WT = littermates, Mann-Whitney-U test, one-tailed, *p = 0.05 [E18.5]). **j** Body weight of control (*n* = 5) and *Crk-SPH* (*n* = 6) pups (control = sgRNA$^{+/+}$ littermates, Mann-Whitney-U test, one-tailed, ns = not significant [E18.5]). **k** Morphology of control and *Crk-SPH* pups; *Crk-SPH* animals appeared cyanotic (control = sgRNA$^{+/+}$ littermates, scale-bar: 1 cm [E18.5]). **l** Percentages of *Crk-SPH* animals at postnatal day 21 (P21) correspond to expected Mendelian ratios (likelihood 12.5%). *Crk-SPH* genotype (sgRNA$^{+/-}$//*dCas9*$^{+/-}$//*Pax7-Cre*$^{+/-}$) is absent at P21 (*n* = 38 control/0 *Crk-SPH*; control = littermates). **m**, **n** GSEA of control (*n* = 3) vs. *Crk-tg* (*n* = 3) quadriceps muscles transcriptome data (WT = littermates, Nominal *p*-value (NOM p-val), *p* = 0.000 (**m**, **n**) [E18.5]). Source data are provided as a Source Data file. Data are presented as mean values +/− SEM in **c**, **d**, **g** and **l**, **j**.

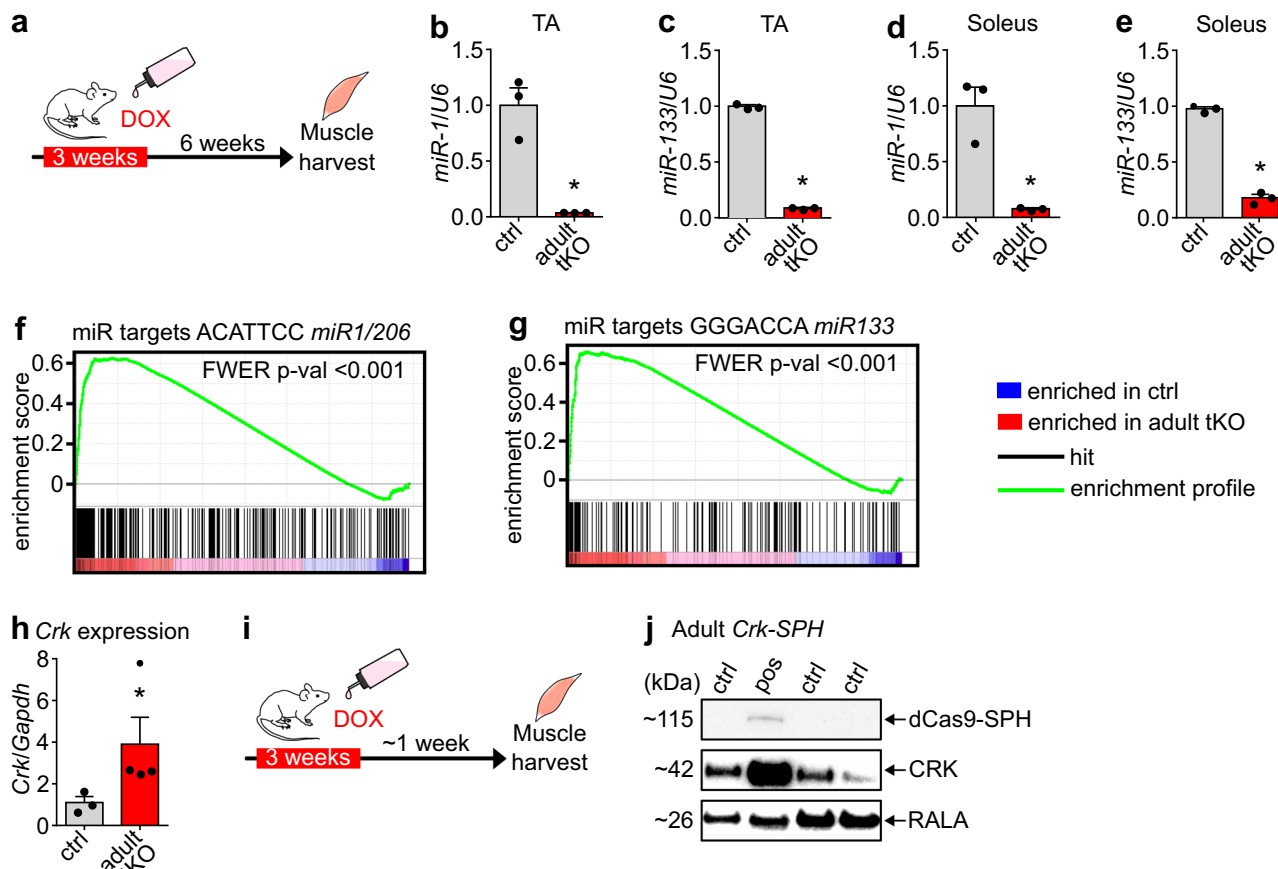

**Fig. 6 Loss of *miR-1/206/133* in adult mice increases CRK levels. a–e** Deletion of *miR-1/133* in adult muscle was induced by doxycycline (DOX) application in *miR-1-1/133a-2*$^{-/-}$//*miR-1-2/133a-1*$^{lox/lox}$//*miR-206/133b*$^{-/-}$//*HSA-rtTA-TRE-Cre*$^{+/-}$ mice (**a**). Doxycycline treatment strongly reduced expression of *miR-1* (**b**, **d**) and *miR-133a* (**c**, **e**) in TA (**b**, **c**) and soleus (**d**, **e**) muscles of control and adult *miR-1/206/133* tKO mice tested via TaqMan assays (*n* = 3 control/3 tKO, control = *miR-1-1/133a-2*$^{+/+}$//*miR-1-2/133a-1*$^{lox/lox}$//*miR-206/133b*$^{+/+}$//*HSA-rtTA-TRE-Cre*$^{+/-}$, Mann-Whitney-U test, one-tailed, *p = 0.05 [male, 15–17 weeks old]). *U6 snRNA* served as control. **f**, **g** GSEA of adult *miR-1/206/133* tKO (*n* = 3) and control (*n* = 2) TA muscles reveals enrichment of predicted *miR-1/206* and *miR-133* targets in adult tKO (control = *miR-1-1/133a-2*$^{-/-}$//*miR-1-2/133a-1*$^{lox/lox}$//*miR-206/133b*$^{-/-}$//*HSA-rtTA-TRE-Cre*$^{+/+}$, familywise-error rate method (FWER), *p* < 0.001 (**f**, **g**) [male, ~44 weeks]). **h** Adult *miR-1/206/133* tKO (*n* = 4) show elevated *Crk* expression tested via TaqMan assay in TA muscles (*n* = 3, control = *miR-1-1/133a-2*$^{-/-}$//*miR-1-2/133a-1*$^{lox/lox}$//*miR-206/133b*$^{-/-}$//*HSA-rtTA-TRE-Cre*$^{+/+}$, Mann-Whitney-U test, one-tailed, *p = 0.0286[male, female, ~44 weeks]). *Gapdh* served as control. **i**, **j** Western blot analysis shows *HSA-rtTA-TRE-Cre* depended dCas9-SPH expression and elevated CRK expression in quadriceps muscle of adult sgRNA$^{+/-}$//*HSA-rtTA-TRE-Cre*$^{+-}$//*CAG-LSL-dCas9-SPH*$^{+/-}$ mice (*n* = 1 adult *Crk-SPH*) (*n* = 3, control= littermates *HSA-rtTA-TRE-Cre*$^{+/+}$ or *CAG-LSL-dCas9-SPH*$^{+/+}$, [male, female; 21 weeks]). RALA served as loading control. Source data are provided as a Source Data file. Data are presented as mean values +/− SEM in **b–e** and **h**.

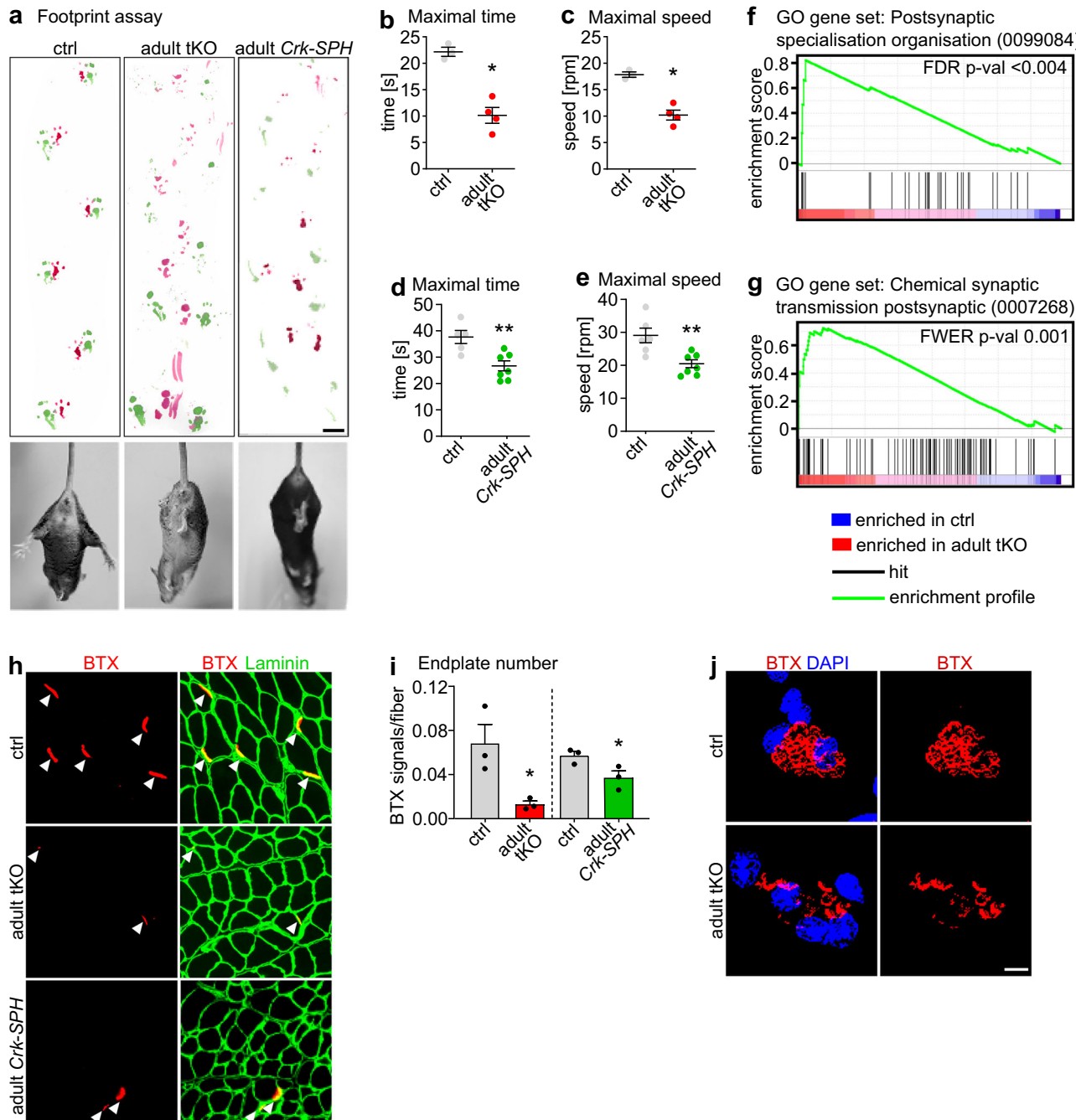

**Fig. 7 Maintenance of NMJ in adult mice requires *miR-1/206/133*-dependent regulation of CRK. a** Adult *miR-1/206/133* tKO (n = 4) and adult *Crk-SPH* (n = 3) show muscle weakness, abnormal locomotion in footprint assays, and hindlimb clasping during tail suspension (n = 3, control = *miR-1-1/133a-2⁻/⁻//miR-1-2/133a-1ˡᵒˣ/ˡᵒˣ//miR-206/133b⁻/⁻//HSA-rtTA-TRE-Cre⁺/⁺* [control and tKO female ~44 weeks; adult *Crk-SPH* 13-16 weeks, females]). Scale-bar: 1 cm. **b, c** Adult *miR-1/206/133* tKO (n = 4) show impaired motor function in Rotarod tests (n = 3, control = *miR-1-1/133a-2⁻/⁻//miR-1-2/133a-1ˡᵒˣ/ˡᵒˣ//miR-206/133b⁻/⁻//HSA-rtTA-TRE-Cre⁺/⁺* [male, female, ~44 weeks]; Mann-Whitney-U test, one-tailed, *p = 0.0286). **d-e** Adult *Crk-SPH* mice show impaired motor functions in Rotarod tests (n = 6, control = *CAG-LSL-dCas9-SPH⁺/⁺*, n = 7 *Crk-SPH* animals [female, 13–16 weeks]; Mann-Whitney-U test, one-tailed, **p = 0.0051 (**d**), **p = 0.0029 (**e**)). **f, g** GSEA of adult *miR-1/206/133* tKO (n = 3) and control (n = 2) TA (control = *miR-1-1/133a-2⁻/⁻//miR-1-2/133a-1ˡᵒˣ/ˡᵒˣ//miR-206/133b⁻/⁻//HSA-rtTA-TRE-Cre⁺/⁺*, familywise-error rate method (FWER), p = < 0.004 (**f**), false discovery rate (FDR) p = 0.001 (**g**) [male, ~44 weeks]). **h, i** Quantification of BTX-stained postsynaptic regions in TA transversal sections from adult *miR-1/206/133* tKO (n = 3) and adult *Crk-SPH* (n = 3) mice in comparison to controls (n = 3, control of adult tKO= *miR-1-1/133a-2⁻/⁻//miR-1-2/133a-1ˡᵒˣ/ˡᵒˣ//miR-206/133b⁻/⁻//HSA-rtTA-TRE- Cre⁺/⁺*, [~44 weeks, male]; n = 3, control of *Crk-SPH = HSA-rtTA-TRE-Cre⁺/⁺* or *CAG-LSL-dCas9-SPH⁺/⁺*[10–14 weeks, female], Mann-Whitney-U tests, one tailed, *p = 0.05). Scale-bar in **r**: 50 μm. **j** BTX staining (red) of isolated single myofibers of *flexor digitorum brevis* muscle from adult *miR-1/206/133* tKO (n = 3) in comparison to controls (n = 3, control of adult *miR-1/206/133* tKO = *miR-1-1/133a-2⁻/⁻//miR-1-2/133a-1ˡᵒˣ/ˡᵒˣ//miR-206/133b⁻/⁻//HSA-rtTA-TRE- Cre⁺/⁺*, [~44 weeks, male]). DAPI staining in blue. Z-stacks. Scale-bar: 20 μm. Source data are provided as a Source Data file. Data are presented as mean values +/− SEM in **b–e** and **i**.

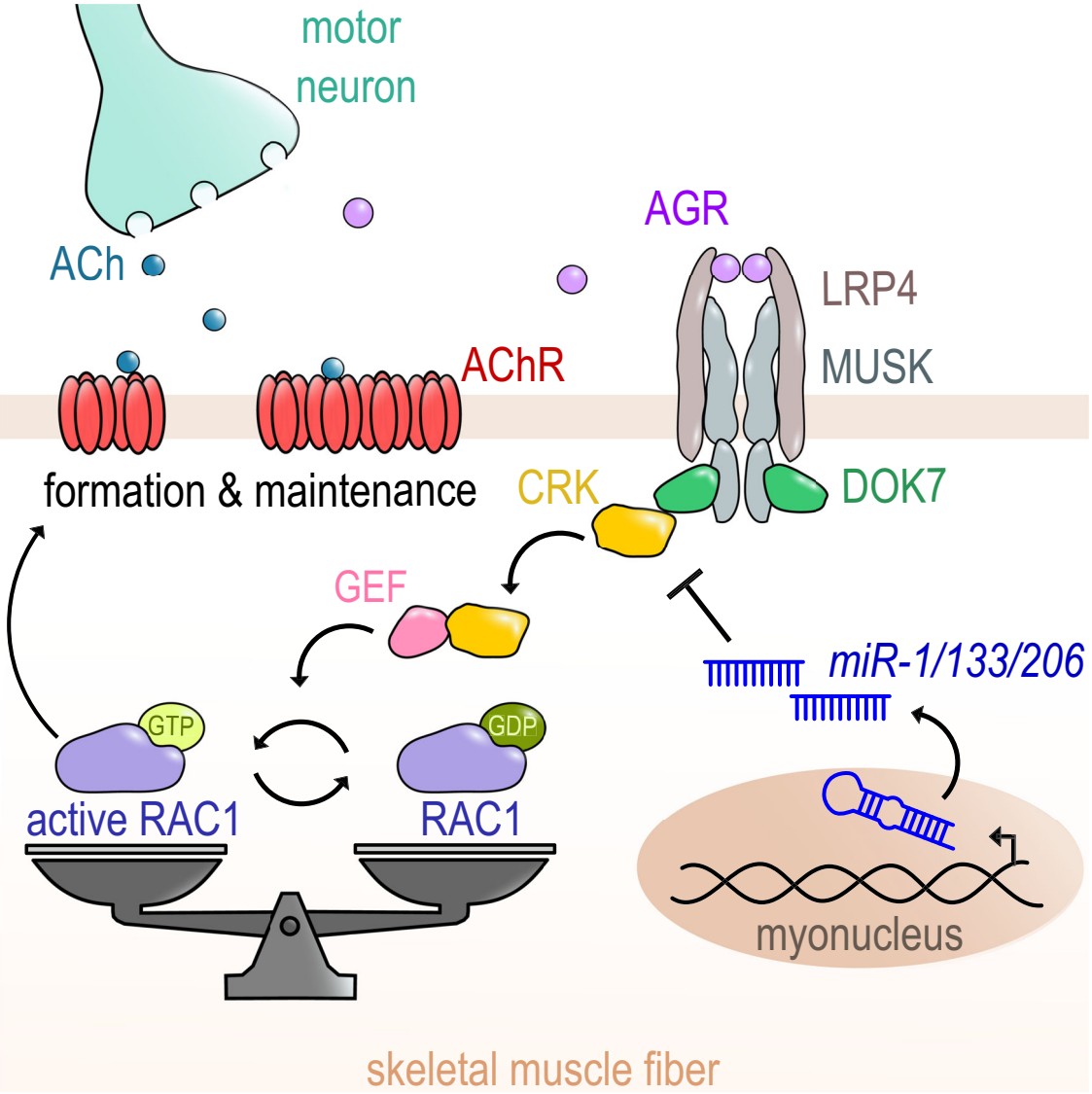

**Fig. 8 Regulation of CRK-RAC1 activity by the *miR-1/206/133* miRNA family is essential for neuromuscular junction function.** The muscle-specific miRNAs of the *miR-1/206/133* family are crucial regulators of a postsynaptic signaling cascade comprising DOK7-CRK-RAC1, critical for stabilization and anchoring of AChRs during NMJ development and maintenance. ACh (Acetylcholine), AChR (Acetylcholine receptor), AGR (agrin), CRK (Adapter protein CRK), DOK7 (Docking Protein 7), GDP (Guanosine diphosphate), GEF (Guanine nucleotide exchange factor), GTP (Guanosine triphosphate), LRP4 (Low-density lipoprotein receptor-related protein 4), MuSK (Muscle-specific kinase), RAC1 (Rac family small GTPase 1).

(synaptic cell adhesion molecule) to promote postsynaptic cytoskeletal dynamics[31]. We demonstrated that enhanced RAC1 activation due to increased CRK expression prevents formation of proper AChR-macro-clusters. Interestingly, tight control of CRK levels is also important to maintain distribution, number, and functionality of NMJs in adult mice. We observed a striking neuromuscular phenotype already one week after completing the 3-weeks DOX treatment to initiate *Crk* overexpression, which argues for an active role of CRK in securing integrity of the NMJ structure. We cannot completely rule out that the increased *Crk* expression interferes with turnover of individual components of the NMJ-complex, but NMJs have been described as relatively stable structures[46], making it unlikely that disturbed turnover is responsible for the rapid onset of the neuromuscular phenotype in adult *Crk-SPH* animals. The late onset of the neuromuscular phenotype in adult *miR-1/206/133* tKO mice is most likely due to slow clearance of highly expressed *miR-1/206/133* miRNAs,

which has been noticed before[14,47]. The rather long time required for downregulation of *miR-1/206/133* miRNA concentrations suggests that these miRNAs are essential to establish long-lasting cellular functions but are not involved in acute regulatory processes.

Adult *miR-1/133a* dKO mice are characterized by upregulation of *Dlk-Dio3* miRNA/lncRNA mega-cluster and mitochondrial dysfunctions[14], a phenotype that was absent in neonatal but not in adult *miR-1/206/133* tKO skeletal muscles. The stage-specific functions of *miR-1/206/133* indicate that miRNAs serve different roles that strongly depend on specific cellular requirements. The same principle was observed for the role of *miR-1/133a* during embryonic heart development, where *miR-1/133a* primarily regulate the switch from primitive smooth muscle gene expressing to more differentiated cardiomyocytes by repressing myocardin and other smooth muscle-specific genes[12]. Another surprising feature of the *miR-1/206/133* family, given the high expression of

individual genes of this family, are concentration-dependent specific functions that manifest as partial redundancy. Adult *miR-1/133a* dKO mice display a mitochondrial but not an NMJ phenotype, whereas adult *miR-1/206/133* tKO animals show both a mitochondrial as well as an NMJ phenotype, although the seed sequences of *miR-1* and *miR-206* or *miR-133a* and *miR-133b* are identical. Absence of a single *miR-206/133b* or *miR-1/133a* miRNA cluster of the *miR-1/206/133* family does not disturb development of the neuromuscular endplate. After loss of two *miR-1/133a* cluster (dKO), the remaining *miR-206/133b* cluster is upregulated, which compensates for the absence of the other clusters in both neonates and adults, enabling normal NMJ development and maintenance. Similarly, inactivation of a single heart-specific *miR-1/133a* cluster is not sufficient to prevent the switch from primitive smooth muscle gene expressing to more differentiated cardiomyocytes[48]. These finding suggest that absolute concentrations of individual miRNAs participate in determining target specificity, probably due the relative efficiency of miRNA-mediated repression for each specific target. Moreover, efficiency of repression may also depend on expression levels of miRNA targets, presence of RNA-binding proteins masking miRNA target sites, regulation of posttranscriptional processing of miRNAs, and RNA-modifications[47,49,50].

The abundance of *miR-1/206/133* and/or their targets may change under disease conditions and exaggerate the role of individual miRNAs of the *miR-1/206/133* family. In a previous study, it was reported that deficiency of *miR-206* alone is sufficient to accelerate progression of ALS[51]. In the same report, it was also claimed that *miR-206* is required for efficient regeneration of neuromuscular synapses, although in a later report from the same group no innervation defects were observed after inactivation of *miR-206* in mdx mice, a commonly used model for muscle regeneration[52]. In contrast, formation of neuromuscular synapses was not affected in *miR-206* mutants[51], which is in line with our results. We assume that expression of *Crk* or other *miR-1/206/133* targets may change under certain disease conditions, which might make the *miR-1/206/133*-CRK-FARP1-RAC1 axis more vulnerable to even small changes in the concentration of *miR-1/206/133* as in *miR-206* knock-out mice.

The stage- and concentration-dependent functions of *miR-1/206/133* create an impressive specificity that allows regulation of a wide range of different developmental and physiological processes according to cellular needs. The regulation of *Crk* represents a paradigmatic example for this principle. CRK is a universal adapter molecule, requiring differential stage-specific regulatory input in different cell types. In myofibers, *miR-1/206/133* have evolved to serve this function and adjust CRK concentrations to support NMJ formation and maintenance. In contrast, cardiomyocytes do not own NMJs and therefore do not require the combined activity of *miR-1/206/133* to regulate *Crk*. It was surprising to learn that *miR-1/206/133* in skeletal muscles are not involved in lineage specification, development, and differentiation as in the heart but primarily direct specific physiological processes. Similarly, *miR-1/206/133* do not play a decisive function in the regulation of MuSCs, which is at odds with previous publications (reviewed in ref.[53]). In this context, it is important to point out that our data rely on clean genetic models avoiding potential off-targets effects that are inevitably associated with the use of antagomirs, GAPmers or other similar techniques[54–56]. On the other hand, germline inactivation of genes may initiate compensatory mechanisms that can disguise specific functions of deleted genes, which may account for differences between miRNA-silencing by antagomirs and genetic approaches. However, we also inactivated *miR-1/206/133* in MuSC and myofibers using inducible Cre-recombinases, essentially eliminating long-term adaptive compensations.

The *miR-1/206/133* target sites in *Crk* are conserved between mice and humans, suggesting that posttranscriptional repression of CRK by *miR-1/206/133* is also operative in other mammalian species. Interestingly, mutations of *Dok7* have been described in human individuals with CMS[6]. The most common mutations of *Dok7* result in C-terminal truncations of DOK7, which may prevent recruitment of CRK and CRK-L, thereby causing NMJ synaptopathy[4]. It seems likely that mutations or processes, which increase the level of CRK account for some cases of idiopathic CMS in humans. Taken together, our findings provide insights into the pathogenesis of human neuromuscular diseases and demonstrate the enormous complexity of regulatory interactions exerted by *miR-1/206/133*.

## Methods

**Study approval**. All animal experiments were done in accordance with the Guide for the Care and Use of Laboratory Animals published by the US National Institutes of Health (NIH Publication No. 85-23, revised 1996) and were approved by the responsible Committee for Animal Rights Protection of the State of Hessen (Regierungspraesidium Darmstadt, Wilhelminenstr. 1-3, 64283 Darmstadt, Germany) with the project number B2-1054 and B2-1234.

**Animal models**. Germline deletion of *miR-1/206/133* (ctKO) was achieved by combining deletion alleles of *miR-1/133a* clusters on mouse chromosome 2 and 18[12] to loss of function alleles of the *miR-206/133b* miRNA cluster[17]. Muscle-specific deletion of the *miR-1/206/133* miRNAs (tKO) was achieved using *Pax7-Cre* mice[21] in combination with *miR-1-1/miR-133a-2−/−//miR-1-2/miR-133a-1lox/lox*[14] and *miR-206/133b−/−* alleles[17]. *miR-1/206/133* were specifically deleted in adult MuSCs (muscle stem cells) by combination of conditional miRNA cluster alleles with *Pax7-CreERT2*[57]. Activation of Cre recombinase activity in adult MuSC was induced by tamoxifen injection (i.p. 75 mg/kg BW/day for 5 days at 8–10 weeks of age). Deletion of *miR-1/206/133* specifically in adult myofibers was achieved using the *HSA-rtTA/TRE-Cre* mouse line[36]. Recombination efficiency in adult myofibers was confirmed by mating of *HSA-rtTA/TRE-Cre* mice to the *Z/AP* reporter strain[37]. Recombination of *Z/AP* or miRNA deletion in adult myofibers was induced by administration of 2 mg/ml doxycycline (Sigma #D9891) with 5% sucrose (Roth #9097.1) in drinking water ad libitum for 3 weeks.

For forced skeletal muscle-specific expression of *Crk*, the ORF of *Crk* was amplified using a DNAbook Fantom2Clone (AK028488, ENSMUST00000017920.13) and oligonucleotides atcgggatcctgATTTCGGGCGCTGCTGGGAGGCTG and cgactggatccCCATCTGTCAGCAAACTGTCGAGC. BamHI digested PCR-fragment was inserted into the pSG5-HSA-MCM vector (kind gift from Karyn Esser;[32]) after deletion of the MCM by EcoRI digestion, relegation and subsequent BamHI digestion. The resulting pSG5-HSA-CRK vector was digested using DrdI and PvuI and the resulting HSA-CRK expression cassette was used for pronuclear injection to generate transgenic mice. Transgenic mice were recovered at E18.5 and presence of the transgene was determined by PCR using transgene-specific oligonucleotides (CCCGAGCCGAG AGTAGCAGTTGTAG, CATCTTCCCATTCTAAACAACACCC).

CRISPR-dCas9-activator-dependent overexpression of *Crk* in skeletal muscle transgenic mice was achieved using specific sgRNA that recruits the dCAS9-SPH complex (CAG-LSL-dCas9-SunTag-p65-HSF1) to the *Crk* promoter after activation of dCAS9-SPH complex expression by *Pax7-Cre* or *HSA-rtTA/TRE-Cre*[33]. Different predicted highly effective sgRNA sequences[34] were used to amplify expression cassettes consisting of U6 promoter, sgRNA and termination sequence using PCR strategy[58]. (sgRNA oligonucleotide #A GTGGAAAGGACGAAAC ACCGGGCCAACCGCGGAGGATGGGGGTTTTAGAGCTAGAAATAG; #B GTGGAAAGGACGAAACACCGGGAGTGCGCATGCGGCGTCCTGTTTTAG AGCTAGAAATAG; #C GTGGAAAGGACGAAACACCGGGGACGCCCGCC ATTGGGAGAGTTTTAGAGCTAGAAATAG; #D GTGGAAAGGACGAAAC ACCGGGAGGCGGAGTCTAGGTCTAAGTTTTAGAGCTAGAAATAG; #E GTGGAAAGGACGAAACACCGGGGACCCCAACCGCGGAGGATGTTTTAG AGCTAGAAATAG). Each generated sgRNA was combined with a U6 promoter derived from plasmid pX459 using PCR. pSpCas9(BB)-2A-Puro (pX459) V2.0 was a gift from Feng Zhang (Addgene plasmid #62988; http://n2t.net/addgene:62988; RRID: Addgene_62988)[59]. Individual expression cassettes were tested for activation of *Crk* transcription in C2 cells in combination with the SP-dCas9-VPR system. SP-dCas9-VPR was a gift from George Church (Addgene plasmid #63798; http://n2t.net/addgene:63798; RRID: Addgene_63798)[35]. Co-transfections were performed using Lipofectamine 3000 (ThermoFisher #L3000075) according to the manufacturer's protocol. The most effective sgRNA cassette (*Crk*-sgRNA #C) was used for pronuclear injection to generate transgenic mice, which used to generate *sgRNA+/−// Pax7-Cre +/−//CAG-LSL-dCas9-SunTag-p65-HSF1+/−* (*Crk-SPH*) and *sgRNA+/−// HSA-rtTA/TRE-Cre +/−//CAG-LSL-dCas9-SunTag-p65-HSF1+/−* (adult *Crk-SPH*) mice. Presence of the sgRNA was confirmed by PCR (GTAAAACGACGGCCAG TGAGGGCCTATTTCCCATGATTC, AGGAAACAGCTATGACCATGAAAAA AAGCACCGACTCGGTGCCAC). (*Pax7-Cre*: GGATAGTGAAACAGGGGCAA, GCTCTGGATACACCTGAGTCT, TCGGCCTTCTTCTAGGTTCTGCTC; *CAG-*

*LSL-dCas9-SPH*: TTCCATTTCAGGTGTCGTGACGTAC, GGTTCTCTTCAG CCGGGTGGCCTCG). sgRNA$^{+/-}$//Pax7-Cre$^{+/-}$//CAG-LSL-dCas9-SunTag-p65-HSF1$^{+/-}$ (*Crk-SPH*) triple heterozygous animals were recovered at E18.5. Triple heterozygous mice for *sgRNA$^{+/-}$//HSA-rtTA/TRE-Cre$^{+/-}$//CAG-LSL-dCas9-SunTag-p65-HSF1$^{+/-}$* (adult *Crk-SPH*; 8-11 weeks old) were treated with 2 mg/ml doxycyclin (Sigma #D9891) in drinking water containing 5% sucrose (Roth #9097.1) ad libitum for 3 weeks. All mice were kept in a C57bl6/129SV mixed background, except for HSA-CRK overexpressing transgenic mice, generated by pronucleus injection, which were in a C57Bl6 background. The sex of mice used for individual experiments is indicated in the respective figure legends. No differences between males and females were evident. The sex of E18.5 fetuses was not determined. All mice were maintained in individually ventilated cages, at 22.5 °C +/− 1 °C and a relative humidity of 50% +/− 5% with controlled illumination (12 h dark/light cycle). Mice were given ad libitum access to food and water.

**Assessment of motor performance in adult mice.** The Rotarod (Panlab) model was used to assess the motor coordination of adult ~44 weeks old mice. Mice were trained bevor performing the test for 1 min at 5 rpm. After passing the trainings phase, four trials were conducted per day at ACC 4 rpm up to 40 rpm in 1 min. Trials were terminated when the animal fell down. For analyses, the average performance was calculated for all four trials of each animal per day. An early tendency to fall down indicated compromised motor coordination. Foot print assays were performed to evaluate motor coordination of ~44 weeks old adult *mir-1/206/133* tKO mice and 13–16 weeks old adult *Crk-SPH* mice. Mouse feet were dipped in nontoxic food color (Wusitta), using a red stain for the fore paws and a green stain for the hind paws, before the mouse was placed in front of a corridor. The animal voluntarily walked along this runway into a hideout, leaving it´s footprints on paper. The assay was repeated two to three times for each mouse. Defects in motor coordination affect gait and footprint pattern. When mice walk in a straight line, the hind paw on one side is typically placed close or into the same spot where the corresponding fore paw has been before.

**Lung floating test.** Assessment of lung expansion in *miR-1/206/133a* tKO mice and their littermates was done at E18.5 following C-section. After removal from the uterus, animals were placed on a preheated plate at 37 °C. Animals were killed 15 min after recovery, lungs were dissected, placed in PBS and the flotation was documented immediately. The genotype of animals was determined afterwards.

**Transmission electron microscopy and Histology.** Quadriceps and diaphragm muscles were collected form pups at E18.5 and fixed with 1.5% paraformaldehyde, 1.5% glutaraldehyde, 0.15 M HEPES at 4 °C for at least 24 h. Fixated samples were subsequently treated with 1% osmium tetroxide solution (Sigma #75633) and uranyl-acetate (Agar Scientific #AGR1260A). Thereafter, the muscle tissue was dehydrated using an ascending series of ethanol. Agar 100 Resin (Agar Scientific #AGR1031) was used for embedding, followed by sectioning using an ultramicrotome. Semi thin longitudinal sections of diaphragm muscles were stained with toluidine blue. Microscopy analysis was performed with a Zeiss LEO 906 and TEM Zeiss EM 902. Histological staining was performed using either sections of paraffin embedded samples or cryosections. For paraffin sections, samples were dehydrated, embedded in paraffin and cut at 8 μm with a Leica #RM125RT microtome. Cryosections were cut at 7–10 μm with Leica CM1950 cryomicrotome and dried at RT for 15 min. H&E staining of sections was performed following standard protocols before embedding in Entellan (Merck #1.07960). Succinate dehydrogenase activity (SDH) staining was performed by staining dried cryosections for 1.5 h at 37 °C with 0.02 M KH$_2$PO$_4$ (Sigma #P5655), 0.076 M Na$_2$HPO$_4$ (Sigma #S3264), 0.1 M sodium succinate, 0.02% w/v nitroblue tetrazolium[14]. LacZ/AP staining[37] was performed with cryosections according to the manufacturer´s protocol (Sigma #F4523).

**Cell culture, transient transfection of cells.** DMEM (Merck #D5796) containing 10% fetal calf serum (Sigma Aldrich #F7524-500ML) and 1% Penicilin-Streptamycin-Glutamine (PSG) was used to keep cells in proliferation. Transient transfections of cell were always performed using Lipofectamine 3000 (Thermo-Fisher #L3000075) according to the manufacturer´s protocol for 24 h or 48 h. For differentiation of C2-cells DMEM containing 2% HS and 1% PSG was used.

**AChR clustering assay.** C2 cells (ATCC #CRL-1772) were seeded in proliferation medium into 96 well plates (Greiner #655090) coated with matrigel (Fisher #CB356238). The medium was switched to differentiation medium at a confluence of ~80% and changed at least every two days. After five to seven days of differentiation transient transfections with different plasmids were performed. The following plasmids were used for transient transfections: pMX GFP RAC$^{G12V}$ (pMX GFP RAC$^{G12V}$ was a gift from Joan Brugge (Addgene plasmid #14567; http://n2t.net/addgene:14567; RRID:Addgene_14567[60]) and pIRESII-EGFP-CRK. The pIRESII-EGFP-CRK vector used for CRK expression under the control of the CMV promoter with GPF reporter was generated by digesting pSG5-HSA-CRK (described above) with BamHI, the ORF of *Crk* was inserted into BamHI site into pIRES2-EGFP (Addgene #6029-1). CRK-L was overexpressed by inserting the *Crk-l* ORF (ENSMUST00000006293.5; bases 393 to 1403) into pIRES2-EGFP (Clontech

#PT3267-5) between the NheI and BamHI sites. The pIRES2-EGFP plasmid served as control plasmid. Myotubes were treated with differentiation medium containing 0.1 μg/ml soluble recombinant agrin protein (R&D Systems #550-AG-100) for 16 h, 48 h after transfection. Cells were fixed at the next day, before staining with the α-Bungarotoxin Alexa Fluor594 conjugate (ThermoFisher #B13423) to label AChRs and with an antibody against GPF as described below.

**Isolation of myofibers.** In brief, the *flexor digitorum brevis* muscle was isolated from the hind limb foot and digested for 2 h using Collagenase P Solution (Sigma #11213873001). Afterwards, the fibers were separated by gently pipetting the digested muscle up and down, followed by transfer into DMEM with 10% FCS (Gibco #10270106) and 1% PSG (ThermoFisher #10378016)[61]. Immunofluorescence staining was performed as described below.

**Luciferase reporter assay.** Quadruplicates of wildtyp and mutated miRNA binding sites were cloned into the pmirGLO Dual-Luciferase Vector (Promega #E1330) regulate expression of firefly luciferase. The vector also contains a renilla luciferase as an internal control. Oligonucleotides were directionally inserted into the NheI and XhoI sites:

*Crk miR-1/206* binding site:
ggauaAUGUAUAUCUUCAUUCCagaggauaAUGUAUAUCUUCAUUCCagaggauaAUGUAUAUCUUCAUUCCagaggauaAUGUAUAUCUUCAUUCCaga;

mutated *Crk miR-1/206* binding site:
ggauaAUGUAUAUCUUGAAACGagaggauaAUGUAUAUCUUGAAACGagaggauaAUGUAUAUCUUGAAACGagaggauaAUGUAUAUCUUGAAACGaga;

*Crk miR-133* binding site:
uCUACUUGGAGUGGAAGGACCAAucuCUACUUGGAGUGGAAGGACCAAucuCUACUUGGAGUGGAAGGACCAAucuCUACUUGGAGUGGAAGGACCAAuc;

mutated *Crk miR-133* binding site:
uCUACUUGGAGUGGAACCAGGAUucuCUACUUGGAGUGGAACCAGGAUucuCUACUUGGAGUGGAACCAGGAUucuCUACUUGGAGUGGAACCAGGAUuc.

Proliferating C2 cells were transfected at 70% confluence with 100 ng DNA/24-well using Lipofectamine 3000 (ThermoFisher #L3000075). Each transfection was performed at least in triplicates. Samples of proliferating C2 cells were collected and lysed after 24 h. For differentiation of C2 cells the medium was switched after transfection to differentiation medium, followed by lysis of transfected cells after one week. Firefly luciferase and Renilla activities were analyzed using the Dual-Luciferase Reporter Assay System (Promega #E1910) and the Mithras LB940 plate reader (Berthold). For normalization the ratio between firefly luciferase intensities to renilla luciferase activities were calculated.

**Immunoprecipitation (IP).** IP was done following established procedure. An extended description is provided in the Supplementary Information file. Protein extracts were prepared using limb muscle tissue (E18.5). The protein lysate was incubated with pre-blocked bead suspension. The supernatants were collected, and antibody was added. Subsequently, the beads were washed, and Co-precipitated proteins were identified by western blot analysis or mass spectrometry.

**Fluorescence activated cell sorting (FACS)-based isolation of MuSC, culture and analysis of confluency.** Isolation and culture of MuSCs was performed by collecting skeletal muscles from individual animals in DMEM (Merck #D5796) containing 2% Penicilin-Streptamycin. Next, the tissue was chopped and digested using Dispase (BD #354-235; 1:10 in solution) and Collagenase II (Worthington #LS004185; 1:10 in solution) for 30 min each. The suspensions were treated with fetal calf serum and filtered through 100 μm, 70 μm and 40 μm cell easystainers. Remaining cells were resuspended after centrifugation and incubated with the following combination of antibodies, anti-CD45-APC, anti-CD31-APC and anti-Ly-6A/E (Sca-1)-APC and anti-CD34-Alexa450 (eBioscience #17-0451, 17-0311, 17-5981, #48-0341-82; each 1:100) and anti-integrin-FITC (MBL #K0046-4; 1:100) for 30 min at 4 °C. Afterwards, the cells were washed, resuspended and incubated with anti-APC MicroBeads (Miltenyi #130-090-855) for 20 min at 4 °C. All CD31, CD45 and Sca-1 positive cells were depleted using the Milteny AutoMACS. The remaining cells were treated with DAPI (Invitrogen, #D1306), to identify dead cells during sorting with a FACS (Aria III, BD FACS Diva v8 Software)[61]. Isolated MuSCs were seeded at a density of 60,000 cells per cm² on matrigel coated dishes in DMEM high glucose GlutaMAX. Half of the medium was renewed every two days. Medium was exchanged to differentiation medium upon confluence. To analyze confluency of isolated MuSCs over time the IncuCyte Zoom system (IncuCyte ZOOM 2016A, Essen Bioscience) was used. MuSCs were cultured in clear bottom 96 well plates (Greiner #655090) as described above, phase contrast images were taken at least every 3 h.

**In situ proximity ligation assay.** In situ proximity ligation assay (PLA) in primary differentiated MuSCs was performed using the DuoLink PLA fluorescence technology (Sigma-Aldrich #DUO92101) according to the manufacturer´s protocol. 200,000 cells were seeded per 96-well. MuSC were fixed after differentiation for 7 min with 4% PFA, followed by permeabilization using 0.3% Triton X-100 in PBS

(3 × 5 min) and two washing steps with PBS. Next, myotubes were incubated with blocking solution for 1 h at 37 °C in a humid chamber. The PLA was done using a combination of two different primary antibodies (anti-CRK Santa Cruz #sc-390132; anti-FARP1 ThermoFisher #PA5-99105), produced either in rabbit or mouse. Antibodies were applied to cells at a dilution of 1:1000 and incubated over night at 4 °C. Following PLA probe incubation, ligation and signal amplification, a Wheat germ agglutinin (WGA) Alexa Fluor 488 Conjugate (ThermoFisher #W11261) staining was performed at a dilution of 1:300 in PBS for 20 min.

**miRNA target prediction analysis.** Potential microRNA target mRNAs were identified using the miRNA target prediction database microrna.org version 2010 (http://www.microrna.org).

**Whole-mount in situ hybridization.** Whole-mount in situ hybridization was performed as described[62]. Briefly, the antisense probe was generated from a plasmid containing a Myogenin cDNA (Source Bioscience clone #IMAGp998A011039Q) with polyA removed by NotI/HpaI digestion and re-ligation using the XhoI line-arized plasmid template in combination with DIG-RNA labelling Mix (Roche #11277073910) and T3 RNA polymerase (Promega #P2083). Unincorporated nucleotides were removed by precipitation of the probe. Embryos of the indicated stage were isolated, fixed using 4% PFA/PBS for 2 h and transferred into PBT (0.1% Tween20/PBS; 2 × 5 min), 25% methanol/PBT, 50% methanol/PBT, 75% methanol/PBT (5 min each step) to methanol (−20 °C, overnight). Samples were processed using 75% methanol/PBT, 50% methanol/PBT, 25% methanol/PBT, 2 x PBT (5 min each) to 6% $H_2O_2$/PBT (1 h, Sigma #H1009). Samples were washed using PBT (3 × 5 min), incubated in 10 μg/ml Proteinase K/PBT (10 min, 21 °C), 2 mg/ml glycine/PBT (5 min), washed using PBT (2 × 5 min) and incubated in hybridization solution (50% Formamide, 5x SSC pH 4.5, 1% SDS, 50 μg/ml yeast tRNA, 50 μg/ml heparin) at 70 °C. After 1 h of prehybridization, probe was added (0.1 μg/ml f. c.) and hybridization was performed overnight at 70 °C. Embryos were transferred to 50% formamide, 5 x SSC pH4.5, 1% SDS (2 × 30 min, 70 °C), 10 mM Tris HCl pH 7.5, 0.5 M NaCl, 0.1% Tween 20 (2 × 5 min, room temperature), 50% formamide, 2 x SSC pH 4.5 (2 × 30 min, 65 °C), MAB solution (3 × 5 min; 100 mM malic acid pH 7.5, 150 mM NaCl), blocking solution (1 h; MAB, 2% blocking reagent [Roche #11096176001], 20% FCS), to anti-Digoxigenin-AP Fab fragments (1:2000, Roche #11093274910 in blocking solution, overnight). Samples were washed using MAB (3 × 5 min, 4 × 1 h), NTMT (3 × 10 min, 100 mM TrisHCl pH 9.5, 50 mM $MgCl_2$; 1% Tween-20; 2 mM Levamisole) and stained in staining solution (21 °C, NBT/BCIP Roche #11681451001 1:50 in NTMT). After staining, embryos were washed in PBT (3 × 5 min) and stored in 50% Glycerol/PBT. All incubations were done at 4 °C unless indicated otherwise.

**Transcriptome analysis and TaqMan assays.** RNA obtained from tissue and cells were isolated using peqGOLD TriFast (VWR #30-2010DE) according to the manufacturer's instructions. RNA was isolated from E18.5 quadriceps muscles and from TA and soleus muscles of adult 12–44 weeks old mice (number of animals are indicated in figure legends). RNA labelling and hybridization to Affymetrix MTA1-0 arrays followed the manufacturer's instructions. Data were analyzed using ArrayStar12 with RMA and evaluated using Student´s t-test. Gene set enrichment analysis (GSEA)[63] was performed using GSEA Java desktop software application v3.0 with default parameters using the permutation type gene set. GSEA determines whether a predefined set of genes shows statistically significant, concordant differences between two biological states and thus helps to identify relevant biological processes[63]. Expression of mature miRNAs and of other transcripts was quantified using FAM labelled TaqMan Gene Expression Assays (ThermoFisher *Crk* #Mm00467065_m1) or TaqMan MicroRNA Assays (ThermoFisher: *miR-1* #002222, *miR-133a* #002246, *miR-206* #000510), respectively. The microRNA Reverse Transcription Kit (#4366596) was used to convert miRNA into cDNA. The TaKaRa Kit (#RR047A) was used for cDNA synthesis of other transcripts. *U6 snRNA* (ThermoFischer #001973) and *Gapdh* (ThermoFischer #Mm999999915) were used as controls. Relative expression was calculated using the ΔΔCt method (StepOne Software v2.3).

**Western blot analysis and immunofluorescence staining.** Western blots and immunofluorescence staining were done following established procedure. An extended description is provided in the Supplementary Information file. Skeletal muscle was lysed by sonication. The protein extracts were separated on a SDS-PAGE and blotted onto nitrocellulose membranes. After the blocking of membranes primary antibodies were used to probe the blots overnight. The secondary antibodies were incubated for 1 h. Quantification of western blots was performed by densitometric scanning.

Cultured cells, isolated myofibers and cryosections were mounted on Superfrost slides and used for immunofluorescence staining. Samples were fixed and washed. Blocking was performed for 1 h. Afterwards the sections were incubated with primary antibodies overnight at 4 °C. Subsequently, the sections were washed and incubated with secondary antibodies and BTX, followed by washing steps. The samples were embedded using Fluoromount W. The Image J software 1.53k was employed for measuring the size of cross-sectional area of Laminin-stained fibers.

Whole-mount staining of the diaphragm followed published protocols[64]. Diaphragm muscles were isolated and stained with α-Bungarotoxin and with antibodies against SV2 and neurofilament. Afterwards, samples were flat-mounted and embedded. The width of AChR endplate zone measured using ImageJ. The AChR cluster size and number was quantified either from serial TA and paraspinal muscle transversal sections or from whole mount diaphragm Z-stack data.

**Pull down assays of active RAC1.** To monitor RAC1 activity after transient overexpression of CRK (pIRESII-EGFP-CRK vector) in C2 cells the Active RAC1 Pull-Down and Detection Kit (ThermoFisher #16118) was used according to the manufacturer's protocol. The kit is based on a GST-fusion protein containing the PBD (p21-binding domain) of PAK1 (human p21-activated protein kinase 1) and glutathione agarose resin to specifically pull down active RAC1. A specific anti-RAC1 antibody was used for detection of RAC1-GTP and total RAC1 on western blot. The western blot procedure was performed as described above. Pull-down assay were performed with samples from three independent transfections per experiment and three independent pull down assays. For analysis, the ratios of RAC1-GTP to total RAC1 were calculated.

**Mass spectrometry (MS).** MS was done following established procedure. An extended description is provided in the Supplementary Information file. Triplicate samples of E18.5 mouse quadriceps where lyzed for whole skeletal muscle proteomics. After determining the concentration of the resulting peptide mixture, equal amounts of peptides where subjected to stable isotope labeling using reductive dimethylation[65,66]. Labeling efficiency was confirmed to be >95% by LC/MS2 and differentially labeled samples mixed 1:1, followed by subsequent desalting using oligo R3 columns. For greater sequencing depth, samples were separated into eight fractions, followed by final desalting, concentration and storage on STAGE tips[67].

Triplicate sample sets of affinity purified CRK from wild type, as well as *miR-1/206/133* tKO E18.5 limb muscle samples were subjected to in gel digest to detect interaction partners of CRK[68]. Processed samples were analyzed using liquid chromatography/tandem-mass spectrome-try (LC/MS2). Relevant instrumentation parameters are included in the supplementary information (Supplementary Data 1)[69]. Peptide/spectrum matching, as well as quantitation was performed using the MaxQuant suite of algorithms (proteome: v. 1.6.0.1; interactome: v. 1.6.6.0[70–72]); against the canonical and isoforms UniProt mouse database (proteome: downloaded 2017/04/20, 97484 entries;interactome: downloaded 2019/08/19, 86161 entries[73]); using parameters documented in the supplementary material (Supplementary Data 1). Downstream bioinformatics analysis was performed using a limma-based R pipeline[74] (https://github.com/bhagwataditya/autonomics).

**Statistics and quantification.** Statistical analysis was performed using GraphPad Prism 6.07. Data are depicted as mean ± SEM. Data values were tested for Gaussian distribution. Students t-test was used as indicated in the figure legends when distribution of data normal. Non-parametric statistical tests were used as indicated, when normality of data distribution could not be confirmed.

**Reporting summary.** Further information on research design is available in the Nature Research Reporting Summary linked to this article.

## Data availability

Data have been deposited in public data bases. Microarray data are available at Arrayexpress (E-MTAB-10229; E-MTAB-10228; E-MTAB-10227). Proteome data are available at ProteomeCentral [http://proteomecentral.proteomexchange.org/cgi/GetDataset?ID=PXD032882] with the dataset identifier (JPST001553) PXD032882. Source data are provided as a Source Data files.

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

## Acknowledgements

We would like to thank Kerstin Richter, Sylvia Thomas, Susanne Kreutzer, Sonja Krueger and Silvia Jeratsch for technical assistance. We thank Ann Atzberger and Johannes Graumann for FACS service and mass spectrometry analysis service, respectively. We thank Ulrich Gärtner and Gerhard Kripp for electron microscopy. This work was supported by the Max-Planck-Society, the Deutsche Forschungsgemeinschaft (D.F.G.) RTG 2355 (T.Boe) and the Transregional Collaborative Research Center TRR267 (T.Boe, T.B.) and TRR81 (T.B.) as well as by the Excellence Cluster Cardio-Pulmonary Institute (C.P.I.) and the German Center for Cardiovascular Research.

## Author contributions

I.K. and T.Boe conceived and designed experiments. I.K. performed most of the experiments, C.S. contributed to experiments on adult mouse motor function and histology, T.G. contributed to experiments on Mu.S.C. differentiation. I.K., T.Boe, and C.S. analyzed the data and prepared figures. T.B. and T.Boe participated in data analysis, discussions and provided advice. I.K., T.Boe, and T.B. wrote the manuscript.

## Funding

## Competing interests

The authors declare no competing interests.
