## [Peer Review File · Nature Communications]

Control of CRK-RAC1 activity by the miR-1/206/133 miRNA family is essential for neuromuscular junction functionREVIEWER COMMENTS

Reviewer #1 (Remarks to the Author):

Klockner et al confirmed that miR-1/206/133 are expressed in skeletal muscles and show similar mature sequence. Triple KO (depletion of miR-1/206/133, tKO) could be obtained at E18.5, but not at P21, suggesting a problem with neonatal survival. tKO was claimed to have no effect on muscle differentiation or formation but reduce the number and size of AChR clusters. They showed that Crk levels were increased in tKO muscles. Forced expression of Crk in C2 muscle cells reduced agrin-induced AChR clusters and in vivo (via transgene injection into embryos) scattered AChR clusters, recapitulating some phenotypes of miR-1/206/133 tKO. They also showed that the interaction of Crk with FARP1 and active RAC1 were increased by tKO. The authors conclude that miR-1/206/133 are not dispensable for NMJ formation, but are dispensable for skeletal muscle development.

miR-1/206/133 miRNAs are a family of miRNAs most abundantly in striated muscles. To this reviewer, it is not unexpected that some NMJ deficits were observed in triple KO mice, which suggests a necessary role. It should be noted that the NMJ phenotypes in tKO mice are minor – AChR clusters are formed, unlike agrin, Lrp4, MuSK, Dok7 or rapsyn mutant mice whose mutation wipes out the NMJ. As evidenced in the literature, there are just too many genes whose mutation may alter a biological event (in this case, NMJ). The key question is whether the miRNAs play an instructive role in a physiological process. This question unfortunately is not addressed for reasons below.

First, this family of miRNAs is abundant in muscles, thus it is extremely important to determine whether tKO alters muscle formation or differentiation. To this reviewer, EM images seemed to suggest a problem in tKO mice: the same-size region contained 9 fibers in control, but only 4-5 fibers in tKO (Figure S3b, enlarged panel). Lower mag images also showed large empty areas between fibers. Images of the diaphragm showed fewer muscle fibers (by a simple count of muscle fibers in cross lines at any given point). Thus, the muscle development seemed to be severely impaired, in contrast to “did not reveal any obvious changes in skeletal morphology”. This raises a concern whether the NMJ deficits are secondary. This question must be addressed by studying more muscles and nicely prepared cross-sections to quantify the number and size of muscle fibers. Because miRNA implication in “oxidative” muscle fibers, the authors should include slow and fast muscles in the study. Finally, muscles should co-stained for nuclei, to determine whether there are regenerated muscle fibers.

Second, is the expression of miR-1/206/133 miRNAs regulated during development, correlating with NMJ formation? In synaptic nuclei? Regulated by agrin/Lrp4/MuSK? What is the physiological context of the proposed model?

Third, the focus on Crk seemed to be narrow and lacked proper controls. Being an abundant miRNA family, miR-1/206/133 tKO is expected to alter the levels of many genes. Readers would benefit to see the list of the target genes. What are they? Figure 3 showed increased protein levels of Crk in tKO mice; how about mRNA levels? In addition, does miR-1/206/133 tKO have any effect on levels of other key proteins in the NMJ formation, including Lrp4, Musk, Dok7, rapsyn and AChR subunits? How about levels of markers of muscle differentiation and regeneration?

Fourth, Crk overexpression (in C2 cells and in embryos) were shown to disrupt AChR clusters; this was claimed via Rac signaling. First of all, it is very difficult to interpret data of overexpression (of any protein), which unavoidably causes non-specific effect. Second, assuming that Crk is an adaptor protein (as proposed in the original paper), overexpressed Crk is likely to disturb the ratio of Crk over its binding partners including MuSK and yet-to-be identified downstream targets.

Finally, thus far, there is no genetic evidence to support the involvement of small G proteins in NMJ formation. Thus, phenotypes obtained by overexpressing Crk did not support that the hypothesis that Crk is downstream of the miRNAs. More convincing evidence would be phenotypes of tKO could be diminished by heterozygous mutation of Crk, which is lacking in the paper.

It is unclear why motor nerves were not characterized in this paper, which could be informative.

Minor concerns:

Line 199 “based on the CRSPR-dCas9-SPH”, should be “CRISPR”.

Line 12 “Formation and maintenance of neuromuscular junctions (NMJs) is essential for skeletal muscle” should be “Formation and maintenance of neuromuscular junctions (NMJs) are essential for skeletal muscle”

"miR-1/-133a-2" or "miR-1/133a-2" should be consistent.

Supplementary figure legend 1 "The miR-206/133b and the miR-1-1/133a-2 clusters were deleted by insertion of a PGK-neomycin selection cassette miR-1-1/133a-2 mutant mice were" should include a full stop after "selection cassette".

Reviewer #2 (Remarks to the Author):

Overall Comments: The manuscript by Klockner and colleagues focuses on the regulation of CRK-RAC1 activity by the myomiR miR-1/206/133 cluster and its essential role in neuromuscular junction (NMJ) function. The authors generated a triple KO (tKO) mouse model and used a doxycycline and tamoxifen-inducible systems to overcome neonatal lethality and study the functions of these miRNAs in regulating key myogenic RNA targets, specifically a transcript (CRK) that affects the NMJ function via RAC1 activation. The authors then use a dCas9 system to drive a Crk transgene under the HSA element which recapitulated the miR-1/206/133 tKO muscle phenotypes. Doxycycline-induced ablation of the miR-1/206/133 myogenic lineage altered the NMJ and requires CRK function.

Overall, this is a well-written manuscript, with a comprehensive set of experiments to elucidate the function of a difficult to study group of muscle microRNAs. There are some questions concerning the timing of regulation of the miRNAs that are being evaluated as well as the phenotypic findings in the miR-1/206/133 muscle tKO mice. If the authors address these concerns, this manuscript could be a very strong addition to the myomiR field of muscle gene expression and regulation.

Major Comments:

1. If elevated CRK protein (resulting from the triple ablation of miR-1/133/206) results in increased activated RAC1 activity, subsequently downstream activated RAC1 targets should also be altered. Filamentous actin should be affected in addition to alterations of AKT/GLUT4 activity. Yet, no alterations in skeletal muscle development/fusion (Supp. Figure 3) appear at least via early myogenic processes.

Have the authors evaluated any of these processes? I was wondering why differentiation and/or fusion of the miR-1/206/133 tKO myoblasts was not affected if RAC1 activity is altered in these mice?

2. There's a large body of work from Thomas Roberts and others showing that circulating miRNAs may both be selectively released from muscle and/or affect distal tissues (Coenen-Stass et al, HMG, 2016; Coenen-Stass et al, Mol. Ther. Nuc. Acids, 2018). Have the authors evaluated the serum levels in the tKO mice to determine if any compensation (e.g. if miRNAs are being activated in other tissues and/or other promoters) to affect either muscle or the NMJ?

3. There are likely additional non-overlapping miRNA targets for each of the miRNAs as they are not expressed at the same levels during myogenic differentiation and development and this point is not discussed (e.g. miR-206 is reported to target Pax7, an MSC marker and affect MSC proliferation and function). Can the authors elaborate more into why and/or how the dysregulation and/or regulation of overlapping and non-overlapping miRNA targets occurs between the clusters? Is there additional post-transcriptional regulation that occurs to regulate the pri-miR transcripts in myofibers and/or the NMJs?

4. Overexpression of just miR-133b and miR-206 have been shown to rescue DMD by the Valdez and Olson labs, which suggests that these miRNAs are uniquely regulated in disease and may be dispensable. The authors should really elaborate on this part and expand on this in the discussion or provide a testable set of experiments to determine if all 3 or just a few are required for proper NMJ formation.

5. Figure 2, for F and G specifically, it's hard to really evaluate the architecture of the NMJ in the tKO mouse diaphragms labeled with bungarotoxin. Can the authors provide a more higher resolution image to confirm the disrupted architecture and NMJ branching that has been reported in the miR-206 KO mouse model (Williams et al., Science, 2009)?

6. Figure 6K, it is not as clear as the authors purported, but there does not appear to be a strong neuromotor defect as other NMJ-defective models. Was clasping of the hindlimbs observed as reported in Alzheimer's and other neuromotor defective mice?

Reviewer #3 (Remarks to the Author):

This paper reports the intriguing finding that mice with a muscle-specific, triple knockout of the miR-1/206/133 family die perinatally, most likely because of respiratory failure. They then go on to provide evidence that this phenotype is based on the posttranscriptional repression of CRK, a phosphorylation-dependent adaptor protein to the MuSK-signaling protein DOK7. Moreover, evidence is provided that CRK binds to FARP1, which in turns is the GEF for Rac1. Overall, I find the work certainly interesting. It appears, however, that the paper is a collections of many data and several aspects that are all not really studied in sufficient depth to make the data compelling. In the following, I will outline the shortcomings. I also would advise the authors to invest time to make the story more coherent in the wording.

Major comments:

1. Description of the NMJ phenotype:

- While I understand that the authors are not experts in the analysis of the NMJ, they need to more fully characterize the NMJ phenotype of the tKO mice. This includes the staining of more muscles. The staining for presynaptic specializations and the motor nerve and the use of whole mount preparations of the muscle to see whether NMJ are innervated or whether there is also a feedback onto the presynaptic side (what would be expected).
- In Figures 2h and i, the authors quantify postsynaptic size and BTX signal. Both of these quantifications can be prone to errors. Muscle fibers have only one single NMJ and they are often localized in a synaptic band. Hence the postsynaptic size when measured in cross-sections is much dependent on where the cross-sections are localized in the synaptic band. How did the authors make sure that they are in comparable localization of the synaptic band in all muscles? In my view, a much better way to determine postsynaptic area would be to use whole-mounts (or longitudinal sections) so that the entire postsynaptic area can be seen. See for example Jones et al 2016; PMID: 5204123 for quantification. A similar argument is valid for the "BTX signal size" displayed in Fig. 2i. Here, it is not at all clear what is actually measured ("%" size range" used to label the x-axis is not at all clear). Please explain what you actually measured. I would also advise to measure this rather in whole mounts.

2. Bioinformatic analysis:

- The presentation and the description of the bioinformatic analyses (Figure 3a-c; Figure 5m, n; Figure 6p, q) is rudimentary and needs to be improved and better explained. Moreover, one of the explanations for the increased expression of synaptic genes might be based on denervation of muscle of tKO mice are denervated. Please discuss this possibility.

3. Expression of miR-1 and miR-133:

- Expression of miR-1 and miR-133 is measured using TaqMan assays. How can one measure miR in HEK cells (a human cell line) and compare this to C2 cells (mouse cell line). Moreover, how is "not expressed" determined (reaching a certain threshold in the number of cycles)? Please explain.

4. Overexpression CRK and Rac1 (fig. 4).

- Transfection using control plasmids is not included. This is an important control as transfection of C2 myotubes with expression plasmids can already affect AChR clustering.
- Is the effect of CRK specific or does overexpression of CRK-L (reported in Hallock et al., 2010; PMID 2964755) also affect AChR clustering?

5. Figure 6

- The result section (p. 9. Line 230 ff) states that there is no expression of miR-1/133a on the adult tKO mice. Again, I wonder how such statement is possible and how expression of miRs in wt mice can be set to 1. In this context, I also wonder whether non-muscle fibers cells in skeletal muscle do not express miRs.

6. FARP1 interaction:

- These data are largely based on proteomics and PLA and I wonder why no additional evidence was shown (eg co-IP of CRK and FARP1).

Detailed response to reviewers

Reviewer #1 (Remarks to the Author):

Klockner et al confirmed that miR-1/206/133 are expressed in skeletal muscles and show similar mature sequence. Triple KO (depletion of miR-1/206/133, tKO) could be obtained at E18.5, but not at P21, suggesting a problem with neonatal survival. tKO was claimed to have no effect on muscle differentiation or formation but reduce the number and size of AChR clusters. They showed that Crk levels were increased in tKO muscles. Forced expression of Crk in C2 muscle cells reduced agrin-induced AChR clusters and in vivo (via transgene injection into embryos) scattered AChR clusters, recapitulating some phenotypes of miR-1/206/133 tKO. They also showed that the interaction of Crk with FARP1 and active RAC1 were increased by tKO. The authors conclude that miR-1/206/133 are not dispensable for NMJ formation, but are dispensable for skeletal muscle development.

miR-1/206/133 miRNAs are a family of miRNAs most abundantly in striated muscles. To this reviewer, it is not unexpected that some NMJ deficits were observed in triple KO mice, which suggests a necessary role. It should be noted that the NMJ phenotypes in tKO mice are minor – AChR clusters are formed, unlike agrin, Lrp4, MuSK, Dok7 or rapsyn mutant mice whose mutation wipes out the NMJ. As evidenced in the literature, there are just too many genes whose mutation may alter a biological event (in this case, NMJ). The key question is whether the miRNAs play an instructive role in a physiological process. This question unfortunately is not addressed for reasons below.

Response: We completely agree with the reviewer that absence of *miR-1/206/133* does not “wipe out” formation of NMJs and that other mutations, e.g. in MuSK, have even stronger effects. However, it is evident that *miR-1/206/133* are critical for generating and maintaining functional NMJs by controlling CRK levels. Absence of *miR-1/206/133* causes perinatal death due to respiratory failure and induces malfunction of NMJs in adult mice, resulting in severe neurological symptoms (paresis). With all due respect, it does not seem adequate to describe perinatal lethality or paresis after loss of *miR-1/206/133* as “minor” phenotypes. Our results reveal that *miR-1/206/133* play an indispensable role for physiological processes, which are formation of functional neuromuscular synapses and their maintenance. As such, we provide compelling evidence that the *miR-1/206/133* – CRK axis plays instructive roles in formation and maintenance of NMJs by adjusting CRK levels.

First, this family of miRNAs is abundant in muscles, thus it is extremely important to determine whether tKO alters muscle formation or differentiation. To this reviewer, EM images seemed to suggest a problem in tKO mice: the same-size region contained 9 fibers in control, but only 4-5 fibers in tKO (Figure S3b, enlarged panel). Lower mag images also showed large empty areas between fibers. Images of the diaphragm showed fewer muscle fibers (by a simple count of muscle fibers in cross lines at any given point). Thus, the muscle development seemed to be severely impaired, in contrast to “did not reveal any obvious changes in skeletal morphology”. This raises a concern whether the NMJ deficits are secondary. This question must be addressed by studying more muscles and nicely prepared cross-sections to quantify the number and size of muscle fibers. Because miRNA implication in “oxidative” muscle fibers, the authors should include slow and fast muscles in the study. Finally, muscles should co-stained for nuclei, to determine whether there are regenerated muscle fibers.

Response: We apologize that the selection of presented images created the impression that muscle development might be impaired, which is not the case. The EM images were included to demonstrate normal structure of sarcomeres but not normal number of muscle fibers. In fact, the original EM figures showed comparable numbers of muscle myofibrils (not fibers) for WT

and *miR-1/133/206* tKO at low magnification (upper part of figure). The “empty” spaces between myofibrils, which are seen both in WT and *miR-1/133/206* tKO, reflect properties of neonatal muscles and often appear when such tissues are prepared for EM. We now provide a different panel of EM images, which represent the absence of morphological changes in *miR-1/133/206* tKO myofibrils in a better way (New Suppl. Fig. 3l).

Furthermore, we have followed the reviewer’s advice and now also include additional cross-sections and quantifications the size and distribution of muscle fibers from different muscles. We did not count the number of fibers, because fiber generation is still ongoing at this time point, making an accurate assessment difficult (see New Suppl. Fig. 3c-d). The new histological data include semi thin sections of the diaphragm (New Suppl. Fig. 3a), a complete overview on H&E-stained hindlimb muscles, comprising both slow and fast muscles (New Suppl. Fig. 3b-c). In addition, we now provide immunofluorescence images of transversal muscle sections and quantifications to demonstrate normal distribution of muscle fibers in in predominantly slow (soleus) and fast (EDL) muscles (New Suppl. Fig. 3d-j). Regenerated fibers cannot be identified in neonatal muscles by central localization of myonuclei, since neonatal muscle fibers do not yet show the typical peripheral localization of myonuclei, characteristic for adult myofibers (New Suppl. figure 3d). To investigate whether muscle fibers in *miR-1/133/206* tKO mice undergo regeneration, we analyzed potential transcriptional changes. We did not observe substantial changes in the expression of muscle development and differentiation marker genes with exception of an upregulation of *Myod1* (New Suppl. Fig. 3k), which is a known consequence of disturbed innervation (Ref: doi: 10.1002/jor.20414). The molecular analysis corroborated our conclusion that *miR-1/133/206* tKO muscles are not undergoing regeneration.

Second, is the expression of miR-1/206/133 miRNAs regulated during development, correlating with NMJ formation? In synaptic nuclei? Regulated by agrin/Lrp4/MuSK? What is the physiological context of the proposed model?

Response: We thank the reviewer for this question. We have included new data describing expression of *miR-1/206/133* miRNAs during development (New Suppl. Fig. 1a-c). *MiR-1/206/133* are the most abundant miRNAs in skeletal muscle based on RNA seq data analysis. We observed robust expression of *miR-1/206/133* at embryonic and fetal stages, which is in line with previous results (doi: 10.1371/journal.pgen.1003793). Interestingly, the expression of *miR-1/206/133* increases in developing skeletal muscles from E14.5 onwards, which correlates with formation of AChR clusters in the middle of myofibers between E16 and P14, following innervation-induced dissemination of non-synaptic clusters (doi: 10.1146/annurev-physiol-022516-034255). Only the expression of *miR-206* declines after P2 (New Suppl. Fig. 1b), since *miR-206* is preferentially found in oxidative muscles. Similar expression profiles of *miR-1/206/133* were also reported for humans (doi: 10.1186/1471-213X-11-34). In our view, the physiological context is evident. We discovered that *miR-1/206/133* play an indispensable role for the physiological formation of functional neuromuscular synapses and their maintenance. We do not know whether *miR-1/206/133* are involved in disease processes affecting the function of NMJs, which is also not the topic of this study.

As recommended by the reviewer, we investigated whether agrin-induced clustering of AChRs has effects on miRNA expression. As shown in (New Suppl. Fig. 8a-c), we did not detect any significant effects of agrin stimulation on the expression of the miRNAs.

The reviewer’s idea about specific expression of miRs in NMJ-associated myonuclei is interesting. Thank you. We re-analyzed publicly available single nuclei RNAseq data sets (doi: 10.1038/s41467-020-18789-8) but did not detect altered expression of miRNA precursor transcripts in NMJ-associated compared to other myonuclei. However, we need to point out

that pri-miRs were only detected in few myonuclei, owing to the rapid processing of pri-miRs to mature miRs.

Third, the focus on Crk seemed to be narrow and lacked proper controls. Being an abundant miRNA family, miR-1/206/133 tKO is expected to alter the levels of many genes. Readers would benefit to see the list of the target genes. What are they? Figure 3 showed increased protein levels of Crk in tKO mice; how about mRNA levels? In addition, does miR-1/206/133 tKO have any effect on levels of other key proteins in the NMJ formation, including Lrp4, Musk, Dok7, rapsyn and AChR subunits? How about levels of markers of muscle differentiation and regeneration?

Response: The focus on *Crk* is not arbitrary but based on the comprehensive analysis of predicted *miR-1/206/133* targets, RNA and protein expression changes, phenotype analysis of *miR-1/206/133* tKO skeletal muscles and functional studies. We favor the hypothesis that functions of a miRNA in a cell or tissue is not only determined by expression level of the respective miRNAs, but also by expression levels of corresponding target genes, presence of RNA-binding proteins, and -most important- the functional significance of target genes, which may vary dependent on the cellular state. Thus, several miRNA target genes may change expression but when a target gene does not play a dominant role, physiological consequences will be minor. Our whole strategy was designed to identify *miR-1/206/133* targets that matter and identify processes that crucially depend on *miR-1/206/133*. *Crk* matches these criteria by all means. Finally, we recapitulated the *miR-1/206/133* tKO phenotype by overexpressing *Crk*, using two different genetic strategies in neonatal as well as in adult mice, which in our view the most stringent control possible (Fig. 5, Fig 6, Fig 7 and Suppl. Fig. 14).

We identified enrichment of multiple predicted targets of *miR-1*, *miR-206* and *miR-133* (Fig 1f-g and Fig. 6f-g). However, detailed analysis of *miR-1/206/133* tKO skeletal muscles revealed that nothing is wrong with myogenesis, sarcomere formation, etc., but that neuromuscular functions were compromised, which brought us to *Crk*. *Crk* was the only gene that fulfilled the criteria for a miRNA target gene regulating NMJ formation (Fig. 3d): (i) *Crk* is a predicted target of *miR-1/206/133*; (ii) *Crk* RNA and CRK protein are upregulated in *miR-1/133/206* tKO muscles (iii) CRK is critical for postsynaptic NMJ formation.

As requested by the reviewer, we extended the description by providing a list of predicted *miR-1/206/133* target genes that are significantly upregulated in *miR-1/133/206* tKO muscles. Postsynaptic NMJ-associated molecules are marked in this list (Suppl. table 2). Please note that the requested data about *Crk* mRNA levels in *miR-1/206/133* tKO muscles were already included in the first version of the manuscript (confirmation of microarray data by RT-qPCR Taqman assay; Fig. 3f). We have also included a list of the expression levels of molecules that are critical for NMJ formation (Suppl. Fig 5g). In line with the previously included gene set enrichment analysis, we observed changes in acetylcholine receptor subunit expression (doi: 10.1016/j.bbrep.2021.101182) as well as upregulation of *Musk* (doi: 10.1007/s13539-011-0041-7.), which corresponds to compromised formation of NMJs, leading to disturbed innervation (see also updated Fig. 2 and New Suppl. Fig. 5).

To better cover potential changes in levels of markers for muscle differentiation and regeneration, we now provide a list of genes involved in muscle differentiation and regeneration (New Suppl. Fig. 3k; Suppl. Fig. 4e). We did not observe substantial changes with exception of an upregulation of *Myod1* (New Suppl. Fig 3k), which is a known consequence of disturbed innervation (Ref: doi: 10.1002/jor.20414).

Fourth, *Crk* overexpression (in C2 cells and in embryos) were shown to disrupt AChR clusters; this was claimed via *Rac* signaling. First of all, it is very difficult to interpret data of overexpression (of any protein), which unavoidably causes non-specific effect. Second, assuming that *Crk* is an adaptor protein (as proposed in the original paper), overexpressed *Crk* is likely to disturb the ratio of *Crk* over its binding partners including MuSK and yet-to-be identified downstream targets. Finally, thus far, there is no genetic evidence to support the involvement of small G proteins in NMJ formation. Thus, phenotypes obtained by overexpressing *Crk* did not support that the hypothesis that *Crk* is downstream of the miRNAs. More convincing evidence would be phenotypes of tKO could be diminished by heterozygous mutation of *Crk*, which is lacking in the paper.

Response: The reviewer is absolutely right that overexpression experiments need to be viewed with caution and might lead to misinterpretations, when not done properly. This is the reason why we took extra care to express *Crk* at only moderate levels and employed two different strategies. The dCAS9-SPH approach increases transcription of the endogenous *Crk* gene, thereby closely mimicking natural processes. Both the transgenic and the dCAS9-SPH approach recapitulated the *miR-1/133/206* tKO phenotype (RNA: compare Fig. 3f to Fig 5g/ Protein: Fig. 3g-h vs Fig. 5b-c), compromising formation and maintenance of NMJ in neonatal and adult muscles. We would also like to point out that even a rather modest increase of CRK expression compromised NMJ formation in skeletal muscles (Fig. 1 for Reviewer 1). Gain-of-function mutations often disrupt physiological processes as rightly pointed out by the reviewer. In fact, the increased expression of *Crk* due to loss of *miR-1/206/133* creates such a gain-of-function condition, in which changes of the ratio of CRK to its binding partners (Suppl. Fig. 6) have detrimental consequences. Furthermore, it was recently reported that CRK does not only interact with DOK7 but also directly with the tyrosine-phosphorylated JM region of MUSK (doi: 10.1038/s41586-021-03672-3). The redundancy in the recruitment of CRK to the synapse may be seen as an indicator for the importance of balanced CRK levels at NMJs. We have added the new citation to the revised manuscript.

Figure 1 for reviewer 1: Western blot analysis of CRK levels in different transgenic founders (Upper panel). Flat-mount preparations of whole diaphragm muscles from different founders compared to littermate (WT) stained with fluorescence-labelled bungarotoxin (lower panel). A, B and C in the upper panel correspond to the correspondingly labelled diaphragms in the lower panel. Please note that even a modest increase of CRK compromised NMJ formation.

It is a reasonable assumption that heterozygosity may lower *Crk* expression, although the outcome may be different in *miR-1/206/133* tKO mice, preventing restoration of appropriate ratios of CRK to its binding partners. We considered such an experiment but eventually discarded it because it is simply too time consuming without any guarantee for success. We would need to generate a combination of a skeletal muscle specific Cre-recombinase, three different homozygous miRNA alleles plus the additional *Crk* mutation, meaning we would deal with eight alleles and a complicated breeding scheme, due to the lethality of individual strains. We sincerely believe that the time and effort required for such an experiment is not justified by the potential gain in knowledge.

The reviewer is right that there is no genetic evidence so far, that small G-proteins are involved in NMJ formation. However, there are reports demonstrating that agrin-induced acetylcholine receptor clustering is mediated by small guanosine triphosphatases of the Rho family e.g. Rac (doi: 10.1083/jcb.150.1.205; doi: 10.1242/jcs.215251). Our data provides further ground to support this hypothesis. Furthermore, we confirmed that expression of a constitutive active version of RAC1 (RAC^{G12V}) impairs AChR-macro-cluster formation (Fig. 4e-f).

It is unclear why motor nerves were not characterized in this paper, which could be informative.

Response: *MiR-1/206/133* is not expressed in motoneurons. Thus, the primary mechanism pertains to the postsynaptic part of the NMJ. Nevertheless, we understand that additional information about the presynaptic part of the NMJ may be interesting the readers. We now include images of control and *miR-1/06/133* tKO whole mount diaphragms stained for SV2 (synaptic vesicles) and neurofilament and BTX to depict axons and nerve terminals from motoneurons (New Fig. 2f, New Suppl. Fig. 5a).

Minor concerns:

Line 199 “based on the CRSPR-dCas9-SPH”, should be “CRISPR”.

Response: We are sorry for this mistake, which has been corrected.

Line 12 “Formation and maintenance of neuromuscular junctions (NMJs) is essential for skeletal muscle” should be “Formation and maintenance of neuromuscular junctions (NMJs) are essential for skeletal muscle”.

Response: Thank you for the careful reading. We have corrected this mistake.

“miR-1-133a-2” or “miR-1/133a-2” should be consistent.

Response: Thanks for the hint. Indeed, a different nomenclature was used in Suppl. Fig. 1 and table 1, which was changed.

Supplementary figure legend 1 “The miR-206/133b and the miR-1-1/133a-2 clusters were deleted by insertion of a PGK-neomycin selection cassette miR-1-1/133a-2 mutant mice were” should include a full stop after “selection cassette”.

Response: Thanks again for the careful reading. The full stop was inserted.

Reviewer #2 (Remarks to the Author):

Overall Comments: The manuscript by Klockner and colleagues focuses on the regulation of CRK-RAC1 activity by the myomiR miR-1/206/133 cluster and its essential role in neuromuscular junction (NMJ) function. The authors generated a triple KO (tKO) mouse model and used a doxycycline and tamoxifen-inducible systems to overcome neonatal lethality and study the functions of these miRNAs in regulating key myogenic RNA targets, specifically a transcript (CRK) that affects the NMJ function via RAC1 activation. The authors then use a dCas9 system to drive a Crk transgene under the HSA element which recapitulated the miR-1/206/133 tKO muscle phenotypes. Doxycycline-induced ablation of the miR-1/206/133 myogenic lineage altered the NMJ and requires CRK function.

Overall, this is a well-written manuscript, with a comprehensive set of experiments to elucidate the function of a difficult to study group of muscle microRNAs. There are some questions concerning the timing of regulation of the miRNAs that are being evaluated as well as the phenotypic findings in the miR-1/206/133 muscle tKO mice. If the authors address these concerns, this manuscript could be a very strong addition to the myomiR field of muscle gene expression and regulation.

Major Comments:

1. If elevated CRK protein (resulting from the triple ablation of miR-1/133/206) results in increased activated RAC1 activity, subsequently downstream activated RAC1 targets should also be altered. Filamentous actin should be affected in addition to alterations of AKT/GLUT4 activity. Yet, no alterations in skeletal muscle development/fusion (Supp. Figure 3) appear at least via early myogenic processes.

Have the authors evaluated any of these processes? I was wondering why differentiation and/or fusion of the miR-1/206/133 tKO myoblasts was not affected if RAC1 activity is altered in these mice?

Response: We thank the reviewer for this insightful comment. Following the reviewer's advice, we performed additional experiments to analyze activation of RAC1 downstream targets. Gene set enrichment analysis (GSEA) of control and *miR-1/206/133* tKO muscles revealed enrichment of the gene sets "Rho GTPase activates PAKs" and "Translocation of SLC2A4 Glut4 to the plasma membrane" in neonatal as well as in adult *miR-1/133/206* tKO muscles (New Suppl. Fig. 7a-b and Suppl. Fig. 11a-b). These findings are in agreement with previous reports (doi: 10.1016/S0960-9822(95)00256-9 & doi: 10.2337/db12-0491). Likewise, forced expression of CRK in adult muscles, stimulating RAC1 activity, increases phosphorylation of PAK1/2/3 (New Suppl. Fig. 11c-e) (doi: 10.4049/jimmunol.171.7.3785; doi: 10.3390/cells8050434). Thus, we can conclude that increased expression of *Crk*, either due to inactivation of *miR-1/206/133* or forced expression of *Crk*, stimulates RAC1 activity, which leads to activation of RAC1 targets.

We found that fusion and differentiation of myoblasts happened normally after inactivation of *miR-1/206/133* (New Suppl. Fig 3 and Suppl. Fig 4), which is indeed puzzling in light of increased RAC1 activity, but may be explained as follows: (i) Our data suggests that the increase of RAC1 activity occur locally, at sites to which CRK is recruited (i.e. forming NMJs). (ii) The increase of RAC1 might be sufficient to disrupt NMJ formation but not sufficiently high to affect filamentous actin and myoblast fusion. Additional mechanisms might compensate changes in RAC activity in regulation of filamentous actin and myoblast fusion.

2. There's a large body of work from Thomas Roberts and others showing that circulating miRNAs may both be selectively released from muscle and/or affect distal tissues (Coenen-Stass et al, HMG, 2016; Coenen-Stass et al, Mol. Ther. Nuc. Acids, 2018). Have the authors evaluated the serum levels in the tKO mice to determine if any compensation (e.g. if miRNAs are being activated in other tissues and/or other promoters) to affect either muscle or the NMJ?

Response: The reviewer is right that there is a large body of literature about circulating miRNAs, which are released by skeletal muscles undergoing turnover but also during myogenic differentiation. Such circulating miRNAs are valuable biomarkers to monitor disease progression in DMD or other muscle diseases. Other reports described effects of miRNAs released by muscle cells on neighboring cells (i.e. acting not via the circulation but rather in a paracrine manner). However, in our study we focus on cell autonomous effect within muscle fibers, where *miR-1/206/133* represent the majority of miRNAs. The only other organ that expresses considerable amounts of *miR-1/133a* but not *miR-206/133b* is the heart. If circulating *miR-1/133a* indeed fulfill a physiological function, it seems possible that release of *miR-1/133a* from cardiomyocytes (or other organs by *de novo* expression) increases in animals lacking *miR-1/133a* in skeletal muscles to maintain constant serum concentration of *miR-1/133a*. It might be interesting to study such phenomena in the future but we think that they are hardly relevant for the current study, in particular since there are no hints for a role of circulating miRNAs for formation or maintenance of NMJs. We now cite corresponding reports but feel that additional experiments into this direction are out of scope for the current study.

3. There are likely additional non-overlapping miRNA targets for each of the miRNAs as they are not expressed at the same levels during myogenic differentiation and development and this point is not discussed (e.g. *miR-206* is reported to target *Pax7*, an MSC marker and affect MSC proliferation and function). Can the authors elaborate more into why and/or how the dysregulation and/or regulation of overlapping and non-overlapping miRNA targets occurs between the clusters? Is there additional post-transcriptional regulation that occurs to regulate the pri-miR transcripts in myofibers and/or the NMJs?

Response: The reviewer rightly points out that there may be non-overlapping targets among some members of the *miR-1/206/133* family. Such non-overlapping targets seem very unlikely for the two *miR-1/133a* clusters, since the mature miRNA sequences are nearly identical. The *miR-206/133b* cluster is a more delicate issue. The sequences between *miR-133a* and *miR-133b* differ only by one base outside the seed sequence, making different targets unlikely as well. In contrast, the sequences of *miR-206* and *miR-1* differ at 4 positions (albeit also outside the seed sequence), enabling targeting of different mRNA molecules. Nevertheless, both *miR-1* and *miR-206* are predicted to control *Crk*, which we confirmed in different experiments. Thus, it seems very likely that we deal with synergistic regulation of *Crk* by *miR-1* and *miR-206*. Of course, this does not exclude differential, at least partially non-overlapping, regulation of other targets by *miR-1* and *miR-206*, which may be due to differences in the sequences and in expression levels. However, our analysis did not disclose that the combined absence of *miR-1* and *miR-206* generates a synthetic phenotype (i.e. a phenotype that arises due to combined dysregulation of different targets), since we were able to mimic the *miR-1/206/133* tKO phenotype by forced expression of *Crk* (Fig. 5 and Fig 7).

Similarly, our data also do not point towards additional post-transcriptional regulation of *miR-1/206/133* pri-miRs. To avoid unsubstantiated speculation, we do not discuss this issue broadly but now mention that regulation of posttranscriptional processing of pri-miRs may affect functions of miRNAs and added an appropriate reference. Differential regulation of targets by miRNAs with identical seed sequences might also be achieved by differential localization of miRNAs within multinucleated myofibers (e.g. differential expression of *miR-1/206/133* genes

in NMJ-associated myonuclei. However, analysis of publicly available single nuclei RNAseq datasets (doi: 10.1038/s41467-020-18789-8), did not reveal any significant differences between NMJ-associated and non-associated myonuclei.

In respect to the regulation of *Pax7* by *miR-206* and potential effects on muscle stem cell proliferation and differentiation. We did not observe any defects in muscle stem cell proliferation and differentiation after inactivation of *miR-1/206/133*, neither *in vivo* nor *in vitro* using isolated primary muscle stem cells. This observation fits to the expression profile of *miR-206*, which is very low in muscle stem cells and only increases during differentiation. Previous reports on the regulation of *Pax7* by *miR-206* were based on rather artificial experimental conditions, which stresses the need to use appropriate *in vivo* models to detect bona fide miRNA targets. We sincerely believe that functions of a miRNA in a cell or tissue is not only determined by expression level of the respective miRNAs, but also by expression levels of corresponding target genes, presence of RNA-binding proteins, and -most important- the functional significance of target genes, which may vary dependent on the cellular state. Obviously, the balance between miRNAs and their targets, as well as the functional relevance of miRNA-targets, often changes under *in vitro* conditions, which might be misleading.

4. *Overexpression of just miR-133b and miR-206 have been shown to rescue DMD by the Valdez and Olson labs, which suggests that these miRNAs are uniquely regulated in disease and may be dispensable. The authors should really elaborate on this part and expand on this in the discussion or provide a testable set of experiments to determine if all 3 or just a few are required for proper NMJ formation.*

Response: We followed the reviewer advice and have expanded the discussion. Our data reveal that inactivation of *miR-1/133a* leads to a substantial up-regulation of *miR-206* (Fig. 1b), which may contribute to the lack of an NMJ-phenotype in *miR-1/133a* dKO mice (Fig. 1c, Fig. 2e, Fig. 7, Suppl. Fig. 12). Likewise, inactivation of the *miR-206/133b* (sKO) gene cluster alone has no effects on NMJ formation and maintenance (Fig. 1c, Fig. 2e), which corresponds to previous reports from our group (doi: 10.1016/j.cmet.2018.02.022, doi: 10.1186/s13395-014-0023-5).

5. *Figure 2, for F and G specifically, it's hard to really evaluate the architecture of the NMJ in the tKO mouse diaphragms labeled with bungarotoxin. Can the authors provide a more higher resolution image to confirm the disrupted architecture and NMJ branching that has been reported in the miR-206 KO mouse model (Williams et al., Science, 2009)?*

Response: We apologize for the low-resolution images of NMJs provided in the initial submission. We now show a more in-depth analysis of NMJs in the neonatal diaphragm as requested (New Fig. 2f and Suppl. Fig. 5b), together with a quantification of reduced AChR cluster numbers and volumes (New Fig. 2g, h). In addition, we would like to point out the structure of neonatal NMJs is rather different compared to adult NMJs, shown in previous publications (doi: 10.1038/35074025; doi: 10.1016/s0092-8674(00)81253-2), including the publication by Williams et al, (2009) (doi: 10.1126/science.1181046). Williams et al. reported that mice deficient for *miR-206* form normal neuromuscular synapses during development. The authors reported that *miR-206* is required for efficient regeneration of neuromuscular synapses after acute nerve injury, which probably accounts for salutary effects in ALS. It was claimed that *miR-206* mediates these effects at least in part through histone deacetylase 4 and fibroblast growth factor signaling pathways. Surprisingly, the same group reported in another publication no innervation defects in *miR-206*-KO;mdx mice compared with mdx mice, which also

undergo continuous re-innervation after regeneration of degenerated myofibers, questioning the previous findings (doi: 10.1172/JCI62656).

We now also provide high-resolution images of pretzel-shaped NMJs from adult control and *miR-1/206/133* tKO muscles, which clearly demonstrate massively altered morphology of NMJs in adult *miR-1/206/133* tKO mutants (New Fig. 7).

6. Figure 6K, it is not as clear as the authors purported, but there does not appear to be a strong neuromotor defect as other NMJ-defective models. Was claspings of the hindlimbs observed as reported in Alzheimer's and other neuromotor defective mice?

Response: We regret if the previous images were misleading. The old figure indeed showed mice with reduced muscle tension and reduced coordination of hindlimbs upon lifting. We now selected different images taken of adult *miR-1/206/133* tKO and adult CRK-SPH mice (New Fig. 7a), which show clear “hind limb claspings” (doi: 10.1186/s40478-016-0377-5; reviewed in doi: 10.1002/0471141755.ph0567s69). The phenotype develops over time in adult mutant mice and becomes more prominent with time. In addition to muscle weakness, hind limb claspings, abnormal footprint pattern and reduced rotarod performance we also observe kyphosis in *miR-1/206/133* tKO mutants (Fig. 1 for reviewer 2).

Figure 1 for reviewer 2: Macroscopic images of adult control, *miR-1/206/133* tKO mutant and *Crk*-overexpressing mice. Adult *miR-1/206/133* tKO mutant mice develop a severe kyphosis after inactivation of *miR-1/206/133* in adult myofibers by administration of doxycycline using *HSA-rtTA/TRE-Cre* mice. The same phenotype is present in *Crk*-overexpressing mice but to a lesser extent.

Reviewer #3 (Remarks to the Author):

This paper reports the intriguing finding that mice with a muscle-specific, triple knockout of the miR-1/206/133 family die perinatally, most likely because of respiratory failure. They then go on to provide evidence that this phenotype is based on the posttranscriptional repression of CRK, a phosphorylation-dependent adaptor protein to the MuSK-signaling protein DOK7. Moreover, evidence is provided that CRK binds to FARP1, which in turns is the GEF for Rac1. Overall, I find the work certainly interesting. It appears, however, that the paper is a collections of many data and several aspects that are all not really studied in sufficient depth to make the data compelling. In the following, I will outline the shortcomings. I also would advise the authors to invest time to make the story more coherent in the wording.

Major comments:

1. Description of the NMJ phenotype:

- While I understand that the authors are not experts in the analysis of the NMJ, they need to more fully characterize the NMJ phenotype of the tKO mice. This includes the staining of more muscles. The staining for presynaptic specializations and the motor nerve and the use of whole mount preparations of the muscle to see whether NMJ are innervated or whether there is also a feedback onto the presynaptic side (what would be expected).

Response: We appreciate that the reviewer finds the study intriguing and interesting. The reviewer is right with the assumption that we have not worked extensively on neuromuscular junction formation before, we are convinced that our findings make a substantial contribution to the field. As correctly pointed out by the reviewer, problems at the postsynaptic side will have secondary consequences for the presynaptic side. Since *miR-1/133/206* are abundantly expressed in muscles but not in nerves, we focused on the postsynaptic side and specifically at the molecular control of signaling processes by *miR-1/133/206* at the postsynaptic part of the NMJ. Nevertheless, we understand that additional information about the presynaptic part of the NMJ may be interesting for the readers. We now include images of control and tKO diaphragms stained for SV2 (synaptic vesicles), neurofilament, and BTX (New Fig. 2f, Suppl. Fig. 5a). The staining corroborates defects in postsynaptic development but also reveals problems at the presynaptic side, as expected by the reviewer.

Motor axons in *miR-1/133/206* tKO muscles are not restricted to a narrow endplate band, but distributed more broadly. We also observed small, not properly innervated AChR clusters, suggesting that some clusters were either never innervated or that motor axons retracted from nascent AChR clusters, which form during the pre-patterning phase (New Fig. 2f, Suppl. Fig. 5a). We assume that the substantial changes in muscle innervation, contributes to muscle weakness and neonatal lethality in *miR-1/133/206* tKO mice.

In addition, we confirm a significant reduction of AChR number and volume employing 3D rendering and visualization of confocal z-stack data in E18.5 whole mount *miR-1/133/206* tKO muscles (New Fig. 2g-h, Suppl. Fig. 5b).

To analyze the mature post-synapse, we examined NMJs in isolated single myofibers from adult *flexor digitorum brevis* muscles. The use of *flexor digitorum brevis* muscles also complies with the reviewer's request to analyze more muscles than just the diaphragm. Consistent with our previous results, we observed fragmented AChR clusters lacking pretzel-shaped branches in adult tKO and adult *Crk*-SPH but not in control animals (New Fig. 7j and Suppl. Fig. 14a). Finally, we analyzed whole mount *extensor digitorum longus* muscles, which essentially display the same phenotypic changes as observed in *flexor digitorum brevis* muscles (New Suppl. Fig. 14b). Unfortunately, we were only able to analyze adult *Crk*-SPH and control mice,

since we initially did not sample *extensor digitorum longus* muscles from adult *miR-1/133/206* tKO mice. Due to the high stability of miRNAs, it takes several months after doxycyclin-initiated deletion of the *miR-1/133/206* genes to sufficiently deplete *miR-1/133/206* levels. An experiment in adult *miR-1/133/206* tKO mice experiment takes >200 days, too long for a revision. In contrast, forced expression of *Crk* causes much more rapid effects, which is the reason why the changes of NMJs in *extensor digitorum longus* muscles were only analyzed in adult *Crk*-SPH mice.

- In Figures 2h and i, the authors quantify postsynaptic size and BTX signal. Both of these quantifications can be prone to errors. Muscle fibers have only one single NMJ and they are often localized in a synaptic band. Hence the postsynaptic size when measured in cross-sections is much dependent on where the cross-sections are localized in the synaptic band. How did the authors make sure that they are in comparable localization of the synaptic band in all muscles? In my view, a much better way to determine postsynaptic area would be to use whole-mounts (or longitudinal sections) so that the entire postsynaptic area can be seen. See for example Jones et al 2016; PMID: 5204123 for quantification. A similar argument is valid for the “BTX signal size” displayed in Fig. 2i. Here, it is not at all clear what is actually measured (“% size range” used to label the x-axis is not at all clear). Please explain what you actually measured. I would also advise to measure this rather in whole mounts.

Response: We agree that quantification of postsynaptic sizes and BTX signals can be prone to error. Of course, we were aware that NMJs are often localized in a synaptic band and therefore adjusted the quantification procedure accordingly. We now provide a more detailed description of the method that was applied. Moreover, we followed the reviewer’s advice and determined postsynaptic areas after whole-mount staining (New Fig. 2g-h, Supp. Fig. 5b).

We could not directly use the method described by Jones et al 2016; PMID: 5204123, since NMJs do not have a pretzel-like morphology at E18.5 but form oval plaques of AChR clusters in the middle region of muscle fibers. Instead, we did a 3D rendering, visualization and quantification of high-magnification, confocal z-stacks from BTX-stained whole mount diaphragms (endplate band region), which confirmed the previous results of an AChR reduction in *miR-1/206/133* tKO. We also revised the labelling of the figures as recommended (now Suppl. Fig. 5e-f).

2. Bioinformatic analysis:

- The presentation and the description of the bioinformatic analyses (Figure 3a-c; Figure 5m, n; Figure 6p, q) is rudimentary and needs to be improved and better explained. Moreover, one of the explanations for the increased expression of synaptic genes might be based on denervation of muscle of tKO mice are denervated. Please discuss this possibility.

Response: We provided a rather short description of the bioinformatic analysis, since the approaches were described in more detail in previous publications. We have introduced an additional reference (doi: 10.1073/pnas.0506580102) and also provide a more extensive and better explanation in the manuscript. The reviewer is right that some of the expression changes of synaptic genes in myofibers may be caused by the reduced innervation of *miR-1/206/133* tKO muscle. We now discuss this possibility in the revised manuscript and also implemented additional data (New Suppl. Fig. 5g).

3. Expression of miR-1 and miR-133:

- Expression of miR-1 and miR-133 is measured using TaqMan assays. How can one measure miR in HEK cells (a human cell line) and compare this to C2 cells (mouse cell line). Moreover, how is “not expressed” determined (reaching a certain threshold in the number of cycles)? Please explain.

Response: We are sorry for the misunderstanding. Human HEK cells were only included as an additional negative control. This is possible because the mature miRNA molecules are identical between mouse and human and thus are detected by the very same Taqman assays. To avoid any confusion, we removed the respective panel from the figure.

4. Overexpression CRK and Rac1 (fig. 4).

- Transfection using control plasmids is not included. This is an important control as transfection of C2 myotubes with expression plasmids can already affect AChR clustering.

- Is the effect of CRK specific or does overexpression of CRK-L (reported in Hallock et al., 2010; PMID 2964755) also affect AChR clustering?

Response: We followed the reviewer’s advice and now show an additional control experiment, in which the overexpression vector without *Crk* insert but with GFP expression was used. Transfection of this control plasmid did not affect AChR clustering (New Suppl. Fig. 8f, g).

We now also performed the AChR clustering assay using a CRK-L overexpression plasmid. Interestingly, overexpression of CRK-L did not affect AChR clustering (New Suppl. Fig. 8d, e). This observation may indicate different properties of CRK versus CRK-L, although redundancy between CRK and CRK-L was reported in loss of function situations. CRK-L is not significantly upregulated in tKO and does not represent a *miR-1/206/133* target gene (Suppl. Fig. 5 g and table 2). Nevertheless, we have included the data into the supplement, since the findings may be interesting to understand potential differences between CRK and CRK-L.

5. Figure 6

- The result section (p. 9. Line 230 ff) states that there is no expression of miR-1/133a on the adult tKO mice. Again, I wonder how such statement is possible and how expression of miRs in wt mice can be set to 1. In this context, I also wonder whether non-muscle fibers cells in skeletal muscle do not express miRs.

Response: We apologize for the misleading wording. The fact that we did detect very little expression of *miR-1/133a* using our assays does not necessarily mean complete absence of expression. We now use the phrase “confirmed a strong reduction...”. To be clear, we did observe residual expression of *miR-1* or *miR-133* using our highly sensitive Taqman assay. Calculations were based on the ratio of *miRNA* to the endogenous control *U6*, which was set to 1 for controls samples. This approach avoids arbitrary *miRNA* values after normalization to *U6* and is commonly used. It may also help to compare relative changes of expression in different experiments.

We cannot completely rule an expression of *miR-1/133/206* in non-muscle cells in skeletal muscles, although it seems clear from numerous studies, including single cell RNA seq experiments, that *miR-1/133/206* are exclusively expressed at noteworthy levels in muscle cells. We detected a very low signal for *miR-1/133a* in neonatal muscles after *Pax7-Cre* mediated deletion of *miR-1/133a* (Suppl. Fig. 1g-i). This might be due to incomplete recombination or expression of *miR-1/133a* in *Pax7*-negative (non-muscle) cells.

Even if there is a very low expression of *miR-1/133a* in non-muscle cells, it would hardly matter for the current study, since the NMJ phenotype was only seen after *Pax7-Cre* mediated deletion and recapitulated after myofiber-specific expression of CRK.

6. *FARP1* interaction:

- *These data are largely based on proteomics and PLA and I wonder why no additional evidence was shown (eg co-IP of CRK and FARP1).*

Response: We are sorry that we did not make this issue clearer. The presented data (now Suppl. Fig. 6c) are based on a Co-IP experiment using a CRK specific antibody and muscle lysates from control and tKO mice (E18.5). Precipitated samples were then analyzed by mass spectrometry. We now show additional data and provide additional information in the figure legend to describe the experiment more accurately (New Suppl. Fig. 6a-b).

To further validate the interaction between CRK and FARP1, we now also show a new co-IP experiment, using an antibody against FARP1 for immunoprecipitation, followed by western blot analysis with a CRK antibody to detect co-immunoprecipitated proteins (Suppl. Fig. 6g).

REVIEWERS' COMMENTS

Reviewer #1 (Remarks to the Author):

The authors have done a pretty good job in addressing the concerns by me and the other two reviewers. I have only a few suggestions on the final figure - the model. For agrin to activate MuSK, two molecules of agrin and two molecules of LRP4 have to form a tetrameric complex (although the agrin-LRP4 dimer forms initially) (see Zong et al., 2012). There is a critical agrin-agrin interface; however, the two agrin molecules are shown far apart. Second, LRP4 should be a transmembrane protein, not as shown as a membrane-attached protein. Third, Dok7 binds to the intracellular justamembrane domain of MuSK, not at the C-terminus as shown. Lin Mei

Reviewer #2 (Remarks to the Author):

The manuscript has been significantly improved since the prior submission. I appreciate the improvements in NMJ imaging (Figure 7) and more detailed gait and clasping phenotypes described in the adult triple KO (tKO) mice. There is likely a significant amount of post-transcriptional regulation of the miR-1/133/206 cluster that also takes place but that would be outside of the scope of this manuscript. There are also likely miRNA/RAC1-independent functions for CRK that might also affect muscle function, but that also is outside of the scope for this particular study. The methodology is sound, and meets the expectations for the muscle biology field. This work will have an impact on the muscle microRNA field and potentially the biomarker field for muscle disease. The results are noteworthy and will likely result in an important impact for the myomiR field. Overall, the manuscript is acceptable and I have no additional comments.

Reviewer #3 (Remarks to the Author):

The authors have done a great job in adding more data. I have no further comments.

Detailed response to the editor's requests

Response: We have modified the manuscript according to the instructions in the Reporting Summary and the Author Checklist. Detailed answers are provided in the Author Checklist.

Detailed response to reviewers

Reviewer #1 (Remarks to the Author):

The authors have done a pretty good job in addressing the concerns by me and the other two reviewers. I have only a few suggestions on the final figure - the model. For agrin to activate MuSK, two molecules of agrin and two molecules of LRP4 have to form a tetrameric complex (although the agrin-LRP4 dimer forms initially) (see Zong et al., 2012). There is a critical agrin-agrin interface; however, the two agrin molecules are shown far apart. Second, LRP4 should be a transmembrane protein, not as shown as a membrane-attached protein. Third, Dok7 binds to the intracellular justamembrane domain of MuSK, not at the C-terminus as shown.

Response: We thank the reviewer for this important contribution. We have followed the reviewer's advice and modified the model shown in Fig. 8. The model now shows that two molecules of agrin and two molecules of LRP4 form a tetrameric complex and that a critical agrin-agrin interface is created when the complex is generated. Furthermore, LRP4 is now correctly shown as a transmembrane protein and the binding of Dok7 to the intracellular justamembrane domain of MuSK is indicated.

Reviewer #2 (Remarks to the Author):

The manuscript has been significantly improved since the prior submission. I appreciate the improvements in NMJ imaging (Figure 7) and more detailed gait and clasping phenotypes described in the adult triple KO (tKO) mice. There is likely a significant amount of post-transcriptional regulation of the miR-1/133/206 cluster that also takes place but that would be outside of the scope of this manuscript. There are also likely miRNA/RAC1-independent functions for CRK that might also affect muscle function, but that also is outside of the scope for this particular study. The methodology is sound, and meets the expectations for the muscle biology field. This work will have an impact on the muscle microRNA field and potentially the biomarker field for muscle disease. The results are noteworthy and will likely result in an important impact for the myomiR field. Overall, the manuscript is acceptable and I have no additional comments.

Response: We are delighted that the reviewer states that we “have significantly improved (the manuscript) since the prior submission”. We agree, it is likely that *miR-1/133/206* cluster does not only regulate CRK levels but is also involved in other post-transcriptional regulatory processes. We have clearly mentioned that *Crk* is most likely not the sole physiologically relevant target of *miR-1/133/206* in skeletal muscle in the paper. On the other hand, our data indicate that the impairment of proper neuromuscular junction formation is the dominant phenotype of *miR-1/133/206* tKO mice and that repression of *Crk* by *miR-1/133/206* plays a major role in neuromuscular junction formation.

Reviewer #3 (Remarks to the Author):

The authors have done a great job in adding more data. I have no further comments.

Response: We appreciate that the reviewer acknowledges “The authors have done a great job in adding more data”.